# Polynomial-Time Linear-Swap Regret Minimization in Imperfect-Information Sequential Games

**Gabriele Farina**
MIT
gfarina@mit.edu

**Charilaos Pipis**
MIT
chpipis@mit.edu

## Abstract

No-regret learners seek to minimize the difference between the loss they cumulated through the actions they played, and the loss they would have cumulated in hindsight had they consistently modified their behavior according to some strategy transformation function. The size of the set of transformations considered by the learner determines a natural notion of rationality. As the set of transformations each learner considers grows, the strategies played by the learners recover more complex game-theoretic equilibria, including correlated equilibria in normal-form games and extensive-form correlated equilibria in extensive-form games. At the extreme, a *no-swap-regret* agent is one that minimizes regret against the set of *all* functions from the set of strategies to itself. While it is known that the no-swap-regret condition can be attained efficiently in nonsequential (normal-form) games, understanding what is the strongest notion of rationality that can be attained *efficiently* in the worst case in *sequential* (extensive-form) games is a longstanding open problem. In this paper we provide a positive result, by showing that it is possible, in any sequential game, to retain polynomial-time (in the game tree size) iterations while achieving sublinear regret with respect to *all linear transformations* of the mixed strategy space, a notion called *no-linear-swap regret*. This notion of hindsight rationality is as strong as no-swap-regret in nonsequential games, and stronger than no-trigger-regret in sequential games—thereby proving the existence of a subset of extensive-form correlated equilibria robust to linear deviations, which we call *linear-deviation correlated equilibria*, that can be approached efficiently.

## 1 Introduction

The framework of regret minimization provides algorithms that players can use to gradually improve their strategies in a repeated game, enabling learning strong strategies even when facing unknown and potentially adversarial opponents. One of the appealing properties of no-regret learning algorithms is that they are *uncoupled*, meaning that each player refines their strategy based on their own payoff function, and on other players' strategies, but not on the payoff functions of other players. Nonetheless, despite their uncoupled nature and focus on *local* optimization of each player's utility, it is one of the most celebrated results in the theory of learning in games that in many cases, when all players are learning using these algorithms, the empirical play recovers appropriate notions of *equilibrium*—a *global* notion of game-theoretic optimality. Strategies constructed via no-regret learning algorithms (or approximations thereof) have been key components in constructing human-level and even superhuman AI agents in a variety of adversarial games, including Poker [Moravčík et al., 2017, Brown and Sandholm, 2018, 2019], Stratego [Perolat et al., 2022], and Diplomacy [Bakhtin et al., 2023].

In regret minimization, each learning agent seeks to minimize the difference between the loss (opposite of reward) they accumulated through the actions they played, and the loss they would have

37th Conference on Neural Information Processing Systems (NeurIPS 2023).

accumulated in hindsight had they consistently modified their behavior according to some strategy transformation function. The size of the set of transformation functions considered by the learning agent determines a natural notion of rationality of the agent. Already when the agents seeks to learn strategies that cumulate low regret against *constant* strategy transformations only—a notion of regret called *external* regret—the average play of the agents converges to a Nash equilibrium in two-player constant-sum games, and to a coarse correlated equilibrium in general-sum multiplayer games. As the sets of transformations the each agent considers grows, more complex equilibria can be achieved, including correlated equilibria in normal-form games (Foster and Vohra [1997], Fudenberg and Levine [1995, 1999], Hart and Mas-Colell [2000, 2001]; see also the monograph by Fudenberg and Levine [1998]) and extensive-form correlated equilibria in extensive-form games [Farina et al., 2022]. At the extreme, a maximally hindsight-rational agent is one that minimizes regret against the set of *all* functions from the strategy space to itself (aka. *swap* regret). While it is known that maximum hindsight rationality can be attained efficiently in nonsequential (normal-form) games [Stoltz and Lugosi, 2007, Blum and Mansour, 2007], it is a major open problem to determine whether the same applies to sequential (*i.e.*, extensive-form) games, and more generally what is the strongest notion of rationality that can be attained efficiently in the worst case in the latter setting.

In this paper, we provide a positive result in that direction, by showing that hindsight rationality can be achieved efficiently in general imperfect-information extensive-form games when one restricts to the set of *all linear transformations* of the mixed strategy space—a notion called *linear-swap regret*, and that coincides with swap regret in normal-form games. In order to establish the result, we introduce several intermediate results related to the geometry of sequence-form strategies in extensive-form games. In particular, a crucial result is given in Theorem 3.1, which shows that the set of linear functions $\mathcal{M}_{\mathcal{Q} \to \mathcal{P}}$ from the sequence-form strategy set $\mathcal{Q}$ of a player in an extensive-form game to a generic convex polytope $\mathcal{P}$ can be captured using only polynomially many linear constraints in the size of the game tree and the number of linear constraints that define $\mathcal{P}$. Applying the result to the special case $\mathcal{P} = \mathcal{Q}$, we are then able to conclude that the the polytope of linear transformations $\mathcal{M}_{\mathcal{Q} \to \mathcal{Q}}$ from the sequence-form strategy set to itself can be captured by polynomially many linear constraints in the size of the game tree, and the norm of any element is polynomially bounded. The polynomial characterization and bound for $\mathcal{M}_{\mathcal{Q} \to \mathcal{Q}}$ is used in conjunction with an idea of Gordon et al. [2008] to construct a no-linear-swap-regret minimizer for the set of strategies $\mathcal{Q}$ starting from two primitives: i) a no-external-regret algorithm for the set of transformations $\mathcal{M}_{\mathcal{Q} \to \mathcal{Q}}$, and ii) an algorithm to compute a fixed point strategy for any transformation in $\mathcal{M}_{\mathcal{Q} \to \mathcal{Q}}$. In both cases, the polynomial representation of $\mathcal{M}_{\mathcal{Q} \to \mathcal{Q}}$ established through Theorem 3.1 plays a fundamental role. It allows, on the one hand, to satisfy requirement ii) using linear programming. On the other hand, it enables us to construct a no-external-regret algorithm that outputs transformations in $\mathcal{M}_{\mathcal{Q} \to \mathcal{Q}}$ with polynomial-time iterations, by leveraging the known properties of online projected gradient descent, exploiting the tractability of projecting onto polynomially-representable polytopes.

Finally, in the last section of the paper we turn our attention away from hindsight rationality to focus instead on the properties of the equilibria that our no-linear-swap-regret dynamics recover in extensive-form games. The average play of no-linear-swap-regret players converges to a set of equilibria that we coin *linear-deviation correlated equilibria (LCEs)*. LCEs form a superset of correlated equilibria and a subset of extensive-form correlated equilibria in extensive-form games. In Section 4 we show that these inclusions are in general strict, and provide additional results about the complexity of computing a welfare-maximizing LCE.

**Related work**   As mentioned in the introduction, the existence of uncoupled no-regret dynamics leading to correlated equilibrium (CE) in multiplayer normal-form games is a celebrated result dating back to at least the work by Foster and Vohra [1997]. That work inspired researchers to seek uncoupled learning procedures in other settings as well. For example, Stoltz and Lugosi [2007] studies learning dynamics leading to CE in games with an infinite (but compact) action set, while Kakade et al. [2003] focuses on graphical games. In more recent years, a growing effort has been spent towards understanding the relationships between no-regret learning and equilibria in imperfect-information extensive-form games, the settings on which we focus. Extensive-form games pose additional challenges when compared to normal-form games, due to their sequential nature and presence of imperfect information. While efficient no-external-regret learning dynamics for extensive-form games are known (including the popular CFR algorithm [Zinkevich et al., 2008]), as of today not much is known about no-swap-regret and the complexity of learning CE in extensive-form games.

In recent work, Anagnostides et al. [2023] construct trigger-regret dynamics that converge to EFCE at a rate of $O(\frac{\log T}{T})$, whereas our regret dynamics converge to linear-deviation correlated equilibria at a slower rate of $O(\frac{1}{\sqrt{T}})$. While the aforementioned paper proposes a general methodology that applies to CEs in normal-form games and EFCE/EFCCE in sequential games, the authors' construction fundamentally relies on being able to express the fixed points computed by the algorithm as (linear combinations of) rational functions with positive coefficients of the deviation matrices. For EFCE and EFCCE this fundamentally follows from the fact that the fixed points can be computed inductively, solving for stationary distributions of local Markov chains at each decision point. The no-linear-swap regret algorithm proposed in our paper does not offer such a local characterization of the fixed point and thus, we cannot immediately transfer the improved regret bounds to our case.

The closest notion to CE that is known to be efficiently computable in extensive-form games is *extensive-form correlated equilibrium* (EFCE) [von Stengel and Forges, 2008, Huang and von Stengel, 2008]. The question of whether the set of EFCE could be approached via uncoupled no-regret dynamics with polynomial-time iterations in the size of the extensive-form games was recently settled in the positive [Farina et al., 2022, Celli et al., 2020]. In particular, Farina et al. [2022] show that EFCE arises from the average play of no-trigger-regret algorithms, where trigger deviations are a particular subset of linear transformations of the sequence-form strategy polytope $\mathcal{Q}$ of each player. Since this paper focuses on learning dynamics that guarantee sublinear regret with respect to *any* linear transformation of $\mathcal{Q}$, it follows immediately that the dynamics presented in this paper recover EFCE as a special case.

The concept of linear-swap-regret minimization has been considered before in the context of *Bayesian* games. Mansour et al. [2022] study a setting where a no-regret *learner* competes in a two-player Bayesian game with a rational utility *maximizer*, that is a strictly more powerful opponent than a learner. Under this setting, it can be shown that in every round the optimizer is guaranteed to obtain at least the Bayesian Stackelberg value of the game. Then they proceed to prove that minimizing *linear-swap regret* is necessary if we want to cap the optimizer's performance at the Stackelberg value, while minimizing polytope-swap regret (a generalization of swap regret for Bayesian games, and strictly stronger than linear-swap) is sufficient to cap the optimizer's performance. Hence, these results highlight the importance of developing learning algorithms under stronger notions of *rationality*, as is our aim in this paper. Furthermore, these results provide evidence that constructing a no-linear-swap regret learner, as is our goal here, can present benefits when compared to other less rational learners. In a concurrent paper, Fujii [2023] defines the notion of *untruthful swap regret* for Bayesian games and proves that, for Bayesian games, it is equivalent to the linear-swap regret which is of interest in our paper.

Bayesian games can be considered as a special case of extensive-form games, where a chance node initially selects one of the possible types $\Theta$ for each player. Thus, our algorithm minimizing linear-swap regret in extensive-form games also minimizes linear-swap regret in Bayesian games. However, we remark that our regret bound depends polynomially on the number of player types $|\Theta|$ as they are part of the game tree representation, while Fujii [2023] has devised an algorithm for Bayesian games, whose regret only depends on $\log|\Theta|$.

Finally, we also mention that nonlinear deviations have been explored in extensive-form games, though we are not aware of notable large sets for which polynomial-time no-regret dynamics can be devised. Specifically, we point to the work by Morrill et al. [2021], which defines the notion of "behavioral deviations". These deviations are nonlinear with respect to the sequence-form representation of strategies in extensive-form games. The authors categorize several known or novel types of restricted behavioral deviations into a Deviation Landscape that highlights the relations between them. Even though both the linear deviations, we consider in this paper, and the behavioral deviations seem to constitute rich measures of rationality, none of them contains the other and thus, linear deviations do not fit into the Deviation Landscape of Morrill et al. [2021] (see also Remark E.2).

## 2    Preliminaries

We recall the standard model of extensive-form games, as well as the framework of learning in games.

## 2.1 Extensive-Form Games

While normal-form games (NFGs) correspond to nonsequential interactions, such as Rock-Paper-Scissors, where players simultaneously pick one action and then receive a payoff based on what others picked, extensive-form games (EFGs) model games that are played on a game tree. They capture both sequential and simultaneous moves, as well as private information and are therefore a very general and expressive model of games, capturing chess, go, poker, sequential auctions, and many other settings as well. We now recall basic properties and notation for EFGs.

**Game tree** In an $n$-player extensive-form game, each node in the game tree is associated with exactly one player from the set $\{1, \ldots, n\} \cup \{c\}$, where the special player $c$—called the *chance* player—is used to model random stochastic outcomes, such as rolling a die or drawing cards from a deck. Edges leaving from a node represent actions that a player can take at that node. To model private information, the game tree is supplemented with an information partition, defined as a partition of nodes into sets called information sets. Each node belongs to exactly one information set, and each information set is a nonempty set of tree nodes for the same Player $i$. An information set for Player $i$ denotes a collection of nodes that Player $i$ cannot distinguish among, given what she has observed so far. (We remark that all nodes in a same information set must have the same set of available actions, or the player would distinguish the nodes). The set of all information sets of Player $i$ is denoted $\mathcal{J}_i$. In this paper, we will only consider *perfect-recall* games, that is, games in which the information sets are arranged in accordance with the fact that no player forgets what the player knew earlier,.

**Sequence-form strategies** Since nodes belonging to the same information set for a player are indistinguishable to that player, the player must play the same strategy at each of the nodes. Hence, a strategy for a player is exactly a mapping from an *information set* to a distribution over actions. In other words, it is the information sets, and not the game tree nodes, that capture the decision points of the player. We can then represent a strategy for a generic player $i$ as a vector indexed by each valid information set-action pair $(j, a)$. Any such valid pair is called a *sequence* of the player; the set of all sequences is denoted as $\Sigma_i := \{(j, a) : j \in \mathcal{J}_i, a \in \mathcal{A}_j\} \cup \{\varnothing\}$, where the special element $\varnothing$ is called *empty sequence*. Given an information set $j \in \mathcal{J}_i$, we denote by $p_j$ the parent sequence of $j$, defined as the last pair $(j, a) \in \Sigma_i$ encountered on the path from the root to any node $v \in j$; if no such pair exists we let $p_j = \varnothing$. Finally, we denote by $\mathcal{C}_\sigma$ the children of sequence $\sigma \in \Sigma_i$, defined as the information sets $j \in \mathcal{J}_i$ for which $p_j = \sigma$. Sequences $\sigma$ for which $\mathcal{C}_\sigma$ is an empty set are called *terminal*; the set of all terminal sequences is denoted $\Sigma_i^\perp$.

**Example 2.1.** *Consider the tree-form decision process faced by Player 1 in the small game of Figure 1 (Left). The decision process has four decision nodes $\mathcal{J}_1 = \{A, B, C, D\}$ and nine sequences including the empty sequence $\varnothing$. For decision node $D$, the parent sequence is $p_D = A2$; for $B$ and $C$ it is $p_B = A1$; for $A$ it is the empty sequence $p_A = \varnothing$.*

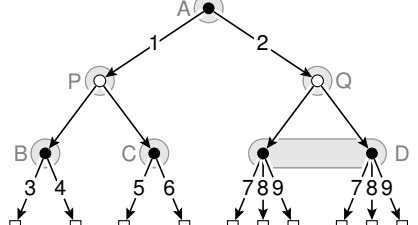

Sequence-form constraints:

$$
\left\{
\begin{aligned}
x[\varnothing] &= 1, \\
x[A1] + x[A2] &= x[\varnothing], \\
x[B3] + x[B4] &= x[A1], \\
x[C5] + x[C6] &= x[A1], \\
x[D7] + x[D8] + x[D9] &= x[A2].
\end{aligned}
\right.
$$

Figure 1: (Left) Tree-form decision process considered in the example. Black round nodes belong to Player 1; white round nodes to Player 2. Square white nodes are terminal nodes in the game tree, payoffs are omitted. Gray bags denote information sets. (Right) The constraints that define the sequence-form polytope $\mathcal{Q}_1$ for Player 1 (besides nonnegativity).

A *reduced-normal-form plan* for Player $i$ represents a deterministic strategy for the player as a vector $x \in \{0, 1\}^{\Sigma_i}$ where the entry corresponding to the generic sequence $x[ja]$ is equal to 1 if the player plays action $a$ at (the nodes of) information set $j \in \mathcal{J}_i$. Information sets that cannot be reached based on the strategy do not have any action select. A crucial property of the reduced-normal-form plan representation of deterministic strategies is the fact that the utility of any player is a multilinear function in the profile of reduced-normal-form plans played by the players. The set of all reduced-

normal-form plans of Player $i$ is denoted with the symbol $\Pi_i$. Typically, the cardinality of $\Pi_i$ is exponential in the size of the game tree.

The convex hull of the set of reduced-normal-form plans of Player $i$ is called the *sequence-form polytope* of the player, and denoted with the symbol $\mathcal{Q}_i := \text{conv}(\Pi_i)$. It represents the set of all randomized strategies in the game. An important result by Romanovskii [1962], Koller et al. [1996], von Stengel [1996] shows that $\mathcal{Q}_i$ can be captured by polynomially many constraints in the size of the game tree, as we recall next.

**Definition 2.2.** *The* polytope of sequence-form strategies *of Player $i$ is equal to the convex polytope*

$$\mathcal{Q}_i := \left\{ \boldsymbol{x} \in \mathbb{R}^\Sigma_{\geq 0} : \begin{array}{ll} (1) & \boldsymbol{x}[\varnothing] = 1 \\ (2) & \sum_{a \in \mathcal{A}_j} \boldsymbol{x}[ja] = \boldsymbol{x}[p_j] \quad \forall\, j \in \mathcal{J} \end{array} \right\}.$$

As an example, the constraints that define the sequence-form polytope for Player 1 in the game of Figure 1 (Left) are shown in Figure 1 (Right). The polytope of sequence-form strategies possesses a strong combinatorial structure that enables speeding up several common optimization procedures and will be crucial in developing efficient algorithms to converge to equilibrium.

## 2.2 Hindsight Rationality and Learning in Games

Games are one of many situations in which a decision-maker has to act in an online manner. For these situations, the most widely used protocol is that of Online Learning (e.g., see Orabona [2022]). Specifically, each learner has a set of actions or behavior they can employ $\mathcal{X} \subseteq \mathbb{R}^d$ (in extensive-form games, this would typically be the set of reduced-normal-form strategies). At each timestep $t$ the learner first selects, possibly at random, an element $\boldsymbol{x} \in \mathcal{X}$, and then receives a loss (opposite of utility) function $\ell^{(t)} : \mathcal{X} \mapsto \mathbb{R}$. Since as we observed the above utilities in EFGs are linear in each player's reduced-normal-form plans, for the rest of the paper we focus on the case in which the loss function $\ell^{(t)}$ is linear, that is, of the form $\ell^{(t)} : \boldsymbol{x} \mapsto \langle \boldsymbol{\ell}^{(t)}, \boldsymbol{x}^{(t)} \rangle$.

A widely adopted objective for the learner is that of ensuring vanishing average *regret* with high probability. Regret is defined as the difference between the loss the learner cumulated through their choice of behavior, and the loss they would have cumulated in hindsight had they consistently modified their behavior according to some strategy transformation function. In particular, let $\Phi$ be a desired set of strategy transformations $\phi : \mathcal{X} \to \mathcal{X}$ that the learner might want to learn not to regret. Then, the learner's $\Phi$-regret is defined as the quantity

$$\Phi\text{-Reg}^{(T)} := \max_{\phi \in \Phi} \sum_{t=1}^{T} \left( \langle \boldsymbol{\ell}^{(t)}, \boldsymbol{x}^{(t)} \rangle - \langle \boldsymbol{\ell}^{(t)}, \phi(\boldsymbol{x}^{(t)}) \rangle \right)$$

A *no-$\Phi$-regret algorithm* (also known as a $\Phi$-regret minimizer) is one that, at all times $T$, guarantees with high probability that $\Phi\text{-Reg}^{(T)} = o(T)$ no matter what is the sequence of losses revealed by the environment. The size of the set $\Phi$ of strategy transformations defines a natural measure of rationality (sometimes called *hindsight rationality*) for players, and several choices have been discussed in the literature. Clearly, as $\Phi$ gets larger, the learner becomes more rational. On the flip side, guaranteeing sublinear regret with respect to all transformations in the chosen set $\Phi$ might be intractable in general. On one end of the spectrum, perhaps the smallest meaningful choice of $\Phi$ is the set of all *constant* transformations $\Phi^{\text{const}} = \{\phi_{\hat{\boldsymbol{x}}} : \boldsymbol{x} \mapsto \hat{\boldsymbol{x}}\}_{\hat{\boldsymbol{x}} \in \mathcal{X}}$. In this case, $\Phi^{\text{const}}$-regret is also called *external regret* and has been extensively studied in the field of online convex optimization. On the other end of the spectrum, *swap regret* corresponds to the setting in which $\Phi$ is the set of *all* transformations $\mathcal{X} \to \mathcal{X}$. Intermediate, and of central importance in this paper, is the notion of *linear-swap regret*, which corresponds to the case in which

$$\Phi := \{\boldsymbol{x} \mapsto \mathbf{A}\boldsymbol{x} : \mathbf{A} \in \mathbb{R}^{d \times d}, \text{ with } \mathbf{A}\boldsymbol{x} \in \mathcal{X} \quad \forall\, \boldsymbol{x} \in \mathcal{X}\} \qquad \text{(linear-swap deviations)}$$

is the set of all linear transformations from $\mathcal{X}$ to itself.[1]

An important observation is that when all considered deviation functions in $\Phi$ are linear, an algorithm guaranteeing sublinear $\Phi$-regret for the set $\mathcal{X}$ can be constructed immediately from a deterministic

---

[1]For the purposes of this paper, the adjective *linear* refers to the fact that each transformation can be expressed in the form $\boldsymbol{x} \mapsto \mathbf{A}\boldsymbol{x}$ for an appropriate matrix $\mathbf{A}$.

no-$\Phi$-regret algorithm for $\mathcal{X}' = \Delta(\mathcal{X})$ by sampling $\mathcal{X} \ni \boldsymbol{x}$ from any $\boldsymbol{x}' \in \mathcal{X}'$ so as to guarantee that $\mathbb{E}[\boldsymbol{x}] = \boldsymbol{x}'$. Since this is exactly the setting we study in this paper, this folklore observation (see also Farina et al. [2022]) enables us to focus on the following problem: does a deterministic no-$\Phi$-regret algorithm for the set of sequence-form strategies $\mathcal{X} = \mathcal{Q}_i$ of any player in an extensive-form game, with guaranteed sublinear $\Phi$-regret in the worst case, exist? In this paper we answer the question for the positive.

**From regret to equilibrium** The setting of Learning in Games refers to the situation in which all players employ a learning algorithm, receiving as loss the negative of the gradient of their own utility evaluated in the strategies output by all the other players. A fascinating aspect of no-$\Phi$-regret learning dynamics is that if each player of a game employs a no-$\Phi$-regret algorithm, then the empirical frequency of play converges almost surely to the set of $\Phi$-equilibria, which are notions of correlated equilibria, in which the rationality of players is bounded by the size of the set $\Phi$. Formally, for a set $\Phi$ of strategy deviations, a $\Phi$-equilibrium is defined as follows.

**Definition 2.3.** *For a $n$-player extensive-form game $G$ and a set $\Phi_i$ of deviations for each player, a $\{\Phi_i\}$-equilibrium is a joint distribution $\mu \in \Delta(\Pi_1 \times \cdots \times \Pi_n)$ such that for each player $i$, and every deviation $\phi \in \Phi_i$ it holds that*

$$\mathbb{E}_{\boldsymbol{x} \sim \mu}[u_i(\boldsymbol{x})] \geq \mathbb{E}_{\boldsymbol{x} \sim \mu}[u_i(\phi(\boldsymbol{x}_i), \boldsymbol{x}_{-i})]$$

*That is, no player $i$ has an incentive to unilaterally deviate from the recommended joint strategy $\boldsymbol{x}$ using any transformation $\phi \in \Phi_i$.*

This general framework captures several important notions of equilibrium across a variety of game theoretic models. For example, in both NFGs and EFGs, no-external regret dynamics converge to the set of Coarse Correlated Equilibria. In NFGs, no-swap regret dynamics converge to the set of Correlated Equilibria [Blum and Mansour, 2007]. In EFGs, Farina et al. [2022] recently proved that a specific subset $\Phi$ of linear transformations called *trigger deviations* lead to the set of EFCE.

**Reducing $\Phi$-regret to external regret** An elegant construction by Gordon et al. [2008] enables constructing no-$\Phi$-regret algorithms for a generic set $\mathcal{X}$ starting from a no-external-regret algorithm for $\Phi$. We briefly recall the result.

**Theorem 2.4** (Gordon et al. [2008]). *Let $\mathcal{R}$ be an external regret minimizer having the set of transformations $\Phi$ as its action space, and achieving sublinear external regret $\mathrm{Reg}^{(T)}$. Additionally, assume that for all $\phi \in \Phi$ there exists a fixed point $\phi(\boldsymbol{x}) = \boldsymbol{x} \in \mathcal{X}$. Then, a $\Phi$-regret minimizer $\mathcal{R}_\Phi$ can be constructed as follows:*

- *To output a strategy $\boldsymbol{x}^{(t)}$ at iteration $t$ of $\mathcal{R}_\Phi$, obtain an output $\phi^{(t)} \in \Phi$ of the external regret minimizer $\mathcal{R}$, and return one of its fixed points $\boldsymbol{x}^{(t)} = \phi^{(t)}(\boldsymbol{x}^{(t)})$.*
- *For every linear loss function $\ell^{(t)}$ received by $\mathcal{R}_\Phi$, construct the linear function $L^{(t)} : \phi \mapsto \ell^{(t)}(\phi(\boldsymbol{x}^{(t)}))$ and pass it as loss to $\mathcal{R}$.*

*Let $\Phi\text{-}\mathrm{Reg}^{(T)}$ be the $\Phi$-regret of $\mathcal{R}_\Phi$. Under the previous construction, it holds that*

$$\Phi\text{-}\mathrm{Reg}^{(T)} = \mathrm{Reg}^{(T)} \qquad \forall\, T = 1, 2, \dots$$

*Thus, if $\mathcal{R}$ is an external regret minimizer then $\mathcal{R}_\Phi$ is a $\Phi$-regret minimizer.*

## 3  A No-Linear-Swap Regret Algorithm with Polynomial-Time Iterations

In this section, we describe our no-linear-swap-regret algorithm for the set of sequence-form strategies $\mathcal{Q}$ of a generic player in any perfect-recall imperfect-information extensive-form game. The algorithm follows the general template for constructing $\Phi$-regret minimizers given by Gordon et al. [2008] and recalled in Theorem 2.4. For this we need two components:

i) an efficient external regret minimizer for the set $\mathcal{M}_{\mathcal{Q} \to \mathcal{Q}}$ of all matrices inducing linear transformations from $\mathcal{Q}$ to $\mathcal{Q}$,

ii) an efficiently computable fixed point oracle for matrices $\mathbf{A} \in \mathcal{M}_{\mathcal{Q} \to \mathcal{Q}}$, returning $\boldsymbol{x} = \mathbf{A}\boldsymbol{x} \in \mathcal{Q}$.

The existence of a fixed point, required in ii), is easy to establish by Brouwer's fixed point theorem, since the polytope of sequence-form strategies is compact and convex, and the continuous function $x \mapsto \mathbf{A}x$ maps $\mathcal{Q}$ to itself by definition. Furthermore, as it will become apparent later in the section, all elements $\mathbf{A} \in \mathcal{M}_{\mathcal{Q} \to \mathcal{Q}}$ have entries in $[0,1]^{\Sigma \times \Sigma}$. Hence, requirement ii) can be satisfied directly by solving the linear feasibility program $\{\text{find } x : \mathbf{A}x = x, x \in \mathcal{Q}\}$ using any of the known polynomial-time algorithms for linear programming. Establishing requirement i) is where the heart of the matter is, and it is the focus of much of the paper. Here, we give intuition for the main insights that contribute to the algorithm. All proofs are deferred to the appendix.

## 3.1 The Structure of Linear Transformations of Sequence-Form Strategy Polytopes

The crucial step in our construction is to establish a series of results shedding light on the fundamental geometry of the set $\mathcal{M}_{\mathcal{Q} \to \mathcal{Q}}$ of *all* linear transformations from a sequence-form polytope $\mathcal{Q}$ to itself. In fact, our results extend beyond functions from $\mathcal{Q}$ to $\mathcal{Q}$ to more general functions from $\mathcal{Q}$ to a generic compact polytope $\mathcal{P} := \{x \in \mathbb{R}^d : \mathbf{P}x = p, x \geq 0\}$ for arbitrary $\mathbf{P}$ and $p$. We establish the following characterization theorem, which shows that when the functions are expressed in matrix form, the set $\mathcal{M}_{\mathcal{Q} \to \mathcal{P}}$ can be captured by a polynomial number of constraints. The proof is deferred to Appendix B.

**Theorem 3.1.** *Let $\mathcal{Q}$ be a sequence-form strategy space and let $\mathcal{P}$ be any bounded polytope of the form $\mathcal{P} := \{x \in \mathbb{R}^d : \mathbf{P}x = p, x \geq 0\} \subseteq [0, \gamma]^d$, where $\mathbf{P} \in \mathbb{R}^{k \times d}$. Then, for any linear function $f : \mathcal{Q} \to \mathcal{P}$, there exists a matrix $\mathbf{A}$ in the polytope*

$$
\mathcal{M}_{\mathcal{Q} \to \mathcal{P}} := \left\{ \mathbf{A} = \left[ \cdots \mid \mathbf{A}_{(\sigma)} \mid \cdots \right] \in \mathbb{R}^{d \times \Sigma} : 
\begin{array}{lll}
(3) & \mathbf{P}\mathbf{A}_{(ja)} = b_j & \forall \, ja \in \Sigma^{\perp} \\
(4) & \mathbf{A}_{(\sigma)} = \mathbf{0} & \forall \, \sigma \in \Sigma \setminus \Sigma^{\perp} \\
(5) & \sum_{j' \in \mathcal{C}_{\varnothing}} b_{j'} = p & \\
(6) & \sum_{j' \in \mathcal{C}_{ja}} b_{j'} = b_j & \forall \, ja \in \Sigma \setminus \Sigma^{\perp} \\
(7) & \mathbf{A}_{(\sigma)} \in [0, \gamma]^d & \forall \, \sigma \in \Sigma \\
(8) & b_j \in \mathbb{R}^k & \forall \, j \in \mathcal{J}
\end{array}
\right\}
$$

*such that $f(x) = \mathbf{A}x$ for all $x \in \mathcal{Q}$. Conversely, any $\mathbf{A} \in \mathcal{M}_{\mathcal{Q} \to \mathcal{P}}$ defines a linear function $x \mapsto \mathbf{A}x$ from $\mathcal{Q}$ to $\mathcal{P}$, that is, such that $\mathbf{A}x \in \mathcal{P}$ for all $x \in \mathcal{Q}$.*

The proof operates by induction in several steps. At its core, it exploits the combinatorial structure of sequence-form strategy polytopes, which can be decomposed into sub-problems using a series of Cartesian products and convex hulls. A high-level description of the induction is as follows:

- The *Base Case* corresponds to the case of being at a leaf decision point. In this case, the set of deviations corresponds to all linear transformations from a probability $n$-simplex into a given polytope $\mathcal{P}$. This set is equivalent to all $d \times n$ matrices whose columns are points in $\mathcal{P}$, which can easily be verified formally. This corresponds to constraint (3) in the characterization of $\mathcal{M}_{\mathcal{Q} \to \mathcal{P}}$.

- For the *Inductive Step*, we are at an intermediate decision point $j$, that is, one for which at least one action leads to further decision points.

  - In Lemma B.8 we show that any terminal action $a$ at $j$ leads to a column in the transformation matrix that is necessarily a valid point in the polytope $\mathcal{P}$. This is similar to the base case, and again leads to constraint (3) in the characterization.

  - In Lemma B.7, we look at the other case of a non-terminal action $a$ at $j$. We prove that there always exists an equivalent transformation matrix whose column corresponding to sequence $ja$ is identically 0 (constraint (4)). This allows for the "crux" of the transformation to happen in the subtrees below $ja$ or equivalently, the subtrees rooted at the children decision points $\mathcal{C}_{ja}$ of $ja$.

  - A key difficulty to conclude the proof is in using the assumption of the inductive step to characterize all such valid transformations. The set of strategies in the subtrees rooted at the children decision points $\mathcal{C}_{ja}$ is in general the Cartesian product of the strategies in these subtrees. This explains the need for the fairly technical Proposition B.4, whose goal is to precisely characterize valid transformations of Cartesian products. This leads to constraints (5) and (6) in our final characterization.

We also remark that while the theorem calls for the polytope $\mathcal{P}$ to be in the form $\mathcal{P} = \{\boldsymbol{x} \in \mathbb{R}^d : \mathbf{P}\boldsymbol{x} = \boldsymbol{p}, \boldsymbol{x} \geq \mathbf{0}\}$, with little work the result can also be extended to handle other representations such as $\{\boldsymbol{x} \in \mathbb{R}^d : \mathbf{P}\boldsymbol{x} \leq \boldsymbol{p}\}$. We opted for the form specified in the theorem since it most directly leads to the proof, and since the constraints that define the sequence-form strategy polytope (Definition 2.2) are already in the form of the statement.

In particular, by setting $\mathcal{P} = \mathcal{Q}$ in Theorem 3.1 (in this case, the dimensions of $\mathbf{P}$ will be $k = |\mathcal{J}| + 1$, and $d = |\Sigma|$), we conclude that the set of linear functions from $\mathcal{Q}$ to itself is a compact and convex polytope $\mathcal{M}_{\mathcal{Q} \to \mathcal{Q}} \subseteq [0, 1]^{\Sigma \times \Sigma}$, defined by $O(|\Sigma|^2)$ linear constraints. As discussed, this polynomial characterization of $\mathcal{M}_{\mathcal{Q} \to \mathcal{Q}}$ is the fundamental insight that enables polynomial-time minimization of linear-swap regret in general extensive-form games.

### 3.2 Our No-Linear-Swap Regret Algorithm

From here, constructing a no-external-regret algorithm for $\mathcal{M}_{\mathcal{Q} \to \mathcal{Q}}$ is relatively straightforward, using standard tools from the rich literature of online learning. For example, in Algorithm 1, we propose a solution employing online projected gradient descent [Gordon, 1999, Zinkevich, 2003].

---

**Algorithm 1:** $\Phi$-Regret minimizer for the set $\Phi = \mathcal{M}_{\mathcal{Q} \to \mathcal{Q}}$

**Data:** $\mathbf{A}^{(1)} \in \mathcal{M}_{\mathcal{Q} \to \mathcal{Q}}$ and fixed point $\boldsymbol{x}^{(1)}$ of $\mathbf{A}^{(1)}$, learning rates $\eta^{(t)} > 0$

1 **for** $t = 1, 2, \ldots$ **do**
2      Output $\boldsymbol{x}^{(t)}$
3      Receive $\boldsymbol{\ell}^{(t)}$ and pay $\langle \boldsymbol{\ell}^{(t)}, \boldsymbol{x}^{(t)} \rangle$
4      Set $\mathbf{L}^{(t)} = \boldsymbol{\ell}^{(t)}(\boldsymbol{x}^{(t)})^\top$
5      $\mathbf{A}^{(t+1)} = \Pi_{\mathcal{M}_{\mathcal{Q} \to \mathcal{Q}}}(\mathbf{A}^{(t)} - \eta^{(t)}\mathbf{L}^{(t)}) = \arg\min_{\mathbf{Y} \in \mathcal{M}_{\mathcal{Q} \to \mathcal{Q}}} \|\mathbf{A}^{(t)} - \eta^{(t)}\mathbf{L}^{(t)} - \mathbf{Y}\|_F^2$
6      Compute a fixed point $\boldsymbol{x}^{(t+1)} = \mathbf{A}^{(t+1)}\boldsymbol{x}^{(t+1)} \in \mathcal{Q}$ of matrix $\mathbf{A}^{(t+1)}$

---

Combining that no-external-regret algorithm for $\Phi$ with the construction by Gordon et al. [2008], we can then establish the following linear-swap regret and iteration complexity bounds for Algorithm 1.

**Theorem 3.2** (Informal). *Let $\Sigma$ denote the set of sequences of the learning player in the extensive-form game, and let $\eta^{(t)} = 1/\sqrt{t}$ for all $t$. Then, for any sequence of loss vectors $\boldsymbol{\ell}^{(t)} \in [0, 1]^\Sigma$, Algorithm 1 guarantees linear-swap regret $O(|\Sigma|^2\sqrt{T})$ after any number $T$ of iterations, and runs in $O(\text{poly}(|\Sigma|)\log^2 t)$ time for each iteration $t$.*

The formal version of the theorem is given in Theorem D.1. It is worth noting that the polynomial-sized description of $\mathcal{M}_{\mathcal{Q} \to \mathcal{Q}}$ is crucial in establishing the polynomial running time of the algorithm, both in the projection step (5) and in the fixed point computation step (6). We also remark that the choice of online projected gradient descent combined with the ellipsoid method for projections were arbitrary and the useful properties of $\mathcal{M}_{\mathcal{Q} \to \mathcal{Q}}$ are retained when using it with any efficient regret minimizer.

## 4 Linear-Deviation Correlated Equilibrium

As we discussed in the preliminaries, when all players in a game employ no-$\Phi$-regret learning algorithms, then the empirical frequency of play converges to the set of $\Phi$-equilibria almost surely. Similarly, when $\Phi = \mathcal{M}_{\mathcal{Q} \to \mathcal{Q}}$ the players act based on "no-linear-swap regret" dynamics and converge to a notion of $\Phi$-equilibrium we call *linear-deviation correlated equilibrium* (LCE). In this section we present some notable properties of the LCE. In particular, we discuss its relation to other already established equilibria, as well as the computational tractability of optimal equilibrium selection.

### 4.1 Relation to CE and EFCE

The $\Phi$-regret minimization framework, offers a natural way to build a hierarchy of the corresponding $\Phi$-equilibria based on the relationship of the $\Phi$ sets of deviations. In particular, if for the sets $\Phi_1, \Phi_2$ it holds that $\Phi_1 \subseteq \Phi_2$, then the set of $\Phi_2$-equilibria is a subset of the set of $\Phi_1$-equilibria. Since

the Correlated Equilibrium is defined using the set of all swap deviations, we conclude that any $\Phi$-equilibrium, including the LCE, is a superset of CE. What is the relationship then of LCE with the extensive-form correlated equilibrium (EFCE)? Farina et al. [2022] showed that the set $\Phi^{\text{EFCE}}$ inducing EFCE is the set of all "trigger deviations", which can be expressed as linear transformations of extensive-form strategies. Consequently, the set $\Phi^{\text{EFCE}}$ is a subset of all linear transformations and thus, it holds that CE $\subseteq$ LCE $\subseteq$ EFCE. In examples E.1 and E.3 of the appendix we show that there exist specific games in which either CE $\neq$ LCE, or LCE $\neq$ EFCE. Hence, we conclude that the previous inclusions are strict and it holds CE $\subsetneq$ LCE $\subsetneq$ EFCE.

For Example E.1 we use a signaling game from von Stengel and Forges [2008] with a known EFCE and we identify a linear transformation that is not captured by the trigger deviations of EFCE. Specifically, it is possible to perform linear transformations on sequences of a subtree based on the strategies on other subtrees of the TFSDP. For Example E.3 we have found a specific game through computational search that has a LCE which is not a normal-form correlated equilibrium. To do that we identify a particular normal-form swap that is non-linear.

**Empirical evaluation**   To further illustrate the separation between no-linear-swap-regret dynamics and no-trigger-regret dynamics, used for EFCE, we provide experimental evidence that minimizing linear-swap-regret also minimizes trigger-regret (Figure 2, left), while minimizing trigger-regret does *not* minimize linear-swap regret. Specifically, in Figure 2 we compare our no-linear-swap-regret learning dynamics (given in Algorithm 1) to the no-trigger-regret algorithm introduced by Farina et al. [2022]. More details about the implementation of the algorithms is available in Appendix F. In the left plot, we measure on the y-axis the average trigger regret incurred when all players use one or the other dynamics. Since trigger deviations are special cases of linear deviations, as expected, we observe that both dynamics are able to minimize trigger regret. Conversely, in the right plot of Figure 2, the y-axis measures linear-swap-regret. We observe that while our dynamics validate the sublinear regret performance proven in Theorem 3.2, the no-trigger-regret dynamics of Farina et al. [2022] exhibit an erratic behavior that is hardly compatible with a vanishing average regret. This suggests that no-linear-swap-regret is indeed a strictly stronger notion of hindsight rationality.

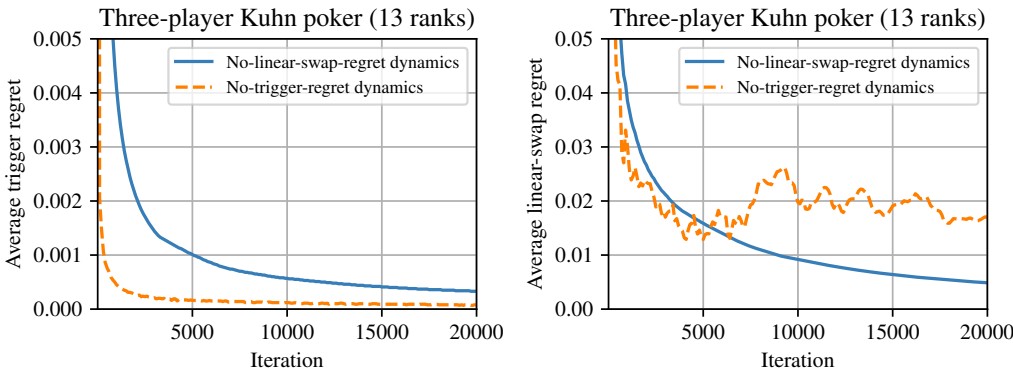

Figure 2: (Left) Average trigger regret per iteration for both a linear-swap-regret minimizer and a trigger-regret minimizer. (Right) Average linear-swap regret per iteration for the same two minimizers.

## 4.2   Hardness of Maximizing Social Welfare

In many cases we are interested in knowing whether it is possible to select an Equilibrium with maximum Social Welfare. Let MAXPAY-LCE be the problem of finding an LCE in EFGs that maximizes the sum (or any linear combination) of all player's utilities. Below, we prove that we cannot efficiently solve MAXPAY-LCE, unless P=NP, even for 2 players if chance moves are allowed, and even for 3 players otherwise. We follow the structure of the same hardness proof for the problem MAXPAY-CE of finding an optimal CE in EFGs. Specifically, von Stengel and Forges [2008] use a reduction from SAT to prove that deciding whether MAXPAY-CE can attain the maximum value is NP-hard even for 2 players. To do that, they devise a way to map any SAT instance into a polynomially large game tree in which the root is the chance player, the second level corresponds to one player, and

the third level corresponds to the other player. The utilities for both players are exactly the same, thus the players will have to coordinate to maximize their payoff irrespective of the linear combination of utilities we aim to maximize.

**Theorem 4.1.** *For two-player, perfect-recall extensive-form games with chance moves, the problem MAXPAY-LCE is not solvable in polynomial time, unless P=NP.*

**Remark 4.2.** *The problem retains its hardness if we remove the chance node and add a third player instead. As showed in von Stengel and Forges [2008], in that case we can always build a polynomially-sized game tree that forces the third player to act as a chance node.*

## 5 Conclusions and Future Work

In this paper we have shown the existence of uncoupled no-linear-swap regret dynamics with polynomial-time iteration complexity in the game tree size in any extensive-form game. This significantly extends prior results related to extensive-form correlated equilibria, and begets learning agents that learn not to regret a significantly larger set of strategy transformations than what was known to be possible before. A crucial technical contribution we made to establish our result, and which might be of independent interest, is providing a polynomial characterization of the set of all linear transformations from a sequence-form strategy polytope to itself. Specifically, we showed that such a set of transformations can be expressed as a convex polytope with a polynomial number of linear constraints, by leveraging the rich combinatorial structure of the sequence-form strategies. Moreover, these no-linear-swap regret dynamics converge to linear-deviation correlated equilibria in extensive-form games, which are a novel type of equilibria that lies strictly between normal-form and extensive-form correlated equilibria.

These new results leave open a few interesting future research directions. Even though we know that there exist polynomial-time uncoupled dynamics converging to linear-deviation correlated equilibrium, we conjecture that it is also possible to obtain an efficient centralized algorithm similar to the Ellipsoid Against Hope for computing EFCE in extensive-form games by Huang and von Stengel [2008]. Additionally, it is an intriguing question to understand whether a no-linear-swap regret algorithm exists that achieves $O(\log T)$ regret per-player, as is the case for no-trigger regret [Anagnostides et al., 2023]. Furthermore, it would be interesting to further explore problems of equilibrium selection related to LCE, possibly by devising suitable Fixed-Parameter Algorithms in the spirit of Zhang et al. [2022]. Finally, the problem of understanding what is the most hindsight rational type of deviations based on which we can construct *efficient* regret minimizers in extensive-form games remains a major open question.

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

# A  Additional Extensive-Form Game Notation

In the proofs, we will make use of the following symbols and notation.

| Symbol | Description |
|---|---|
| $\mathcal{J}_i$ | Set of all Player $i$'s infosets. |
| $\mathcal{A}_j$ | Set of actions available at any node in the information set $j$. |
| $\Sigma_i$ | Set of sequences for Player $i$, defined as $\Sigma^{(i)} := \{(j,a) : j \in \mathcal{J}, a \in A_j\} \cup \{\varnothing\}$, |
| $\varnothing$ | where the special element $\varnothing$ is called the *empty sequence*. |
| $\Sigma_i^\perp$ | Set of terminal sequences for Player $i$. |
| $p_j$ | Parent sequence of $j$, defined as the last pair $(j,a) \in \Sigma_i$ encountered on the path from the root to any information set $j$. |
| $\mathcal{C}_\sigma$ | Set of all "children" of sequence $\sigma$, defined as the information sets $j \in \mathcal{J}$ having as parent $p_j = \sigma$. |
| $j' \prec j$ | Information set $j \in \mathcal{J}$ is an ancestor of $j' \in \mathcal{J}$, that is, there exists a path in the game tree connecting a node $h \in j$ to some node $h' \in j'$. |
| $\sigma \prec \sigma'$ | Sequence $\sigma$ precedes sequence $\sigma'$, where $\sigma, \sigma'$ belong to the same player. |
| $\sigma \succeq j$ | Sequence $\sigma = (j', a')$ is such that $j' \succeq j$. |
| $\Sigma_{\succeq j}$ | Sequences at $j \in \mathcal{J}$ and all of its descendants, $\Sigma_{\succeq j} := \{\sigma \in \Sigma : \sigma \succeq j\}$. |
| $\mathcal{Q}_i$ | Sequence-form strategies of Player $i$ (Definition 2.2). |
| $\mathcal{Q}_{\succeq j}$ | Sequence-form strategies for the subtree rooted at $j \in \mathcal{J}$ (Definition A.1). |
| $\Pi_i$ | Reduced-normal-form plans (a.k.a. deterministic sequence-form strategies) of Player $i$. |
| $\Pi_{\succeq j}$ | Reduced-normal-form plans (a.k.a. deterministic sequence-form strategies) for the subtree rooted at $j \in \mathcal{J}$. |

Table 1: Summary of game-theoretic notation used in this paper. Note that we might skip player-specific subscripts when they can be inferred.

Furthermore, when the subscript referring to players can be inferred or is irrelevant (that is, the quantities refer to a generic player), then we might skip it.

As hinted by some of the rows in the above table, we will sometimes find it important to consider partial strategies that only specify behavior at a decision node $j$ and all of its descendants $j' \succ j$. We make this formal through the following definition.

**Definition A.1.** *The set of sequence-form strategies for the subtree rooted at $j$, denoted $\mathcal{Q}_{\succeq j}$, is the set of all vectors $\boldsymbol{x} \in \mathbb{R}_{\geq 0}^{\Sigma_{\succeq j}}$ such that probability-mass-conservation constraints hold at decision node $j$ and all of its descendants $j' \succ j$, specifically*

$$\mathcal{Q}_{\succeq j} := \left\{ \boldsymbol{x} \in \mathbb{R}_{\geq 0}^{\Sigma_{\succeq j}} : \begin{array}{ll} (9) & \sum_{a \in \mathcal{A}_j} \boldsymbol{x}[ja] = 1 \\ (10) & \sum_{a \in \mathcal{A}_{j'}} \boldsymbol{x}[j'a] = \boldsymbol{x}[p_{j'}] \quad \forall j' \succ j \end{array} \right\}.$$

Finally, we define the symbol $\Sigma_i^\perp$ to be the set of all terminal sequences for Player $i$. Thus, the set $\Sigma_i \setminus \Sigma_i^\perp$ would give us all the non-terminal sequences of that player.

**Access to coordinates**   By definition, sequence-form strategies are vectors indexed by sequences. To access the coordinate corresponding to sequence $\sigma$, we will use the notation $\boldsymbol{x}[\sigma]$. Occasionally, we will need to extract a subvector corresponding to all sequences that are successor of an information set $j$, that is, all sequences $\sigma \succeq j$. For that, we use the notation $\boldsymbol{x}[\succeq j]$.

**Remark on the structure of sequence-form strategies**   We further remark the following known fact about the structure of sequence-form strategies. Intuitively, it crystallizes the idea that sequence-form strategies encode product of probabilities of actions on the path from the root to any decision point. The proof follows directly from the definitions.

**Lemma A.2.** *Let $j \in \mathcal{J}_i$ be an information set for a generic player. Then, given any sequence-form strategy $\boldsymbol{x} \in \mathcal{Q}_{\succeq j}$, action $a \in \mathcal{A}_j$, and child information set $j' \in \mathcal{C}_{ja}$, there exists a sequence-form strategy $\boldsymbol{x}_{\succeq j'} \in \mathcal{Q}_{\succeq j'}$ such that*

$$\boldsymbol{x}[\succeq j'] = \boldsymbol{x}[ja]\boldsymbol{x}_{\succeq j'}.$$

# B  Proof of the Characterization Theorem (Theorem 3.1)

In this section we prove the central result of this paper, the characterization given in Theorem 3.1 of linear functions from the sequence-form strategy polytope $\mathcal{Q}$ to the generic polytope $\mathcal{P} := \{\boldsymbol{x} \in \mathbb{R}^d : \mathbf{P}\boldsymbol{x} = \boldsymbol{p}, \boldsymbol{x} \geq \boldsymbol{0}\}$, where $\mathbf{P} \in \mathbb{R}^{k \times d}$ and $\boldsymbol{p} \in \mathbb{R}^k$.

We will prove the characterization theorem by induction on the structure of the extensive-form strategy polytope. To do so, it will be useful to introduce a few additional objects and notations. We do so in the next subsection.

## B.1  Additional Objects and Notation Used in the Proof

First, we introduce a parametric version of the polytope $\mathcal{P}$, where the right-hand side vector is made variable.

**Definition B.1.** *Given any $\boldsymbol{b} \in \mathbb{R}^k$, we will denote with $\mathcal{P}(\boldsymbol{b})$ the polytope*

$$\mathcal{P}(\boldsymbol{b}) := \{\boldsymbol{x} \in \mathbb{R}^d : \mathbf{P}\boldsymbol{x} = \boldsymbol{b}, \boldsymbol{x} \geq \boldsymbol{0}\}.$$

*In particular, $\mathcal{P} = \mathcal{P}(\boldsymbol{p})$.*

Furthermore, we introduce the equivalence relation $\cong_{\mathcal{D}}$ to indicate that two matrices induce the same linear function when restricted to domain $\mathcal{D}$.

**Definition B.2.** *Given two matrices $\mathbf{A}, \mathbf{B}$ of the same dimension, we write $\mathbf{A} \cong_{\mathcal{D}} \mathbf{B}$ if $\mathbf{A}\boldsymbol{x} = \mathbf{B}\boldsymbol{x}$ for all $\boldsymbol{x} \in \mathcal{D}$. Similarly, given two sets $\mathcal{U}, \mathcal{V}$ of matrices we write $\mathcal{U} \cong_{\mathcal{D}} \mathcal{V}$ to mean that for any $\mathbf{A} \in \mathcal{U}$ there exists $\mathbf{B} \in \mathcal{V}$ with $\mathbf{A} \cong_{\mathcal{D}} \mathbf{B}$, and vice versa.*

Additionally, we introduce a symbol to denote the set of all matrices that induce linear functions from a set $\mathcal{U}$ to a set $\mathcal{V}$.

**Definition B.3.** *Given any sets $\mathcal{U}, \mathcal{V}$ we denote with $\mathcal{L}_{\mathcal{U} \to \mathcal{V}}$ the set of all matrices that induce linear transformations from $\mathcal{U}$ to $\mathcal{V}$, that is,*

$$\mathcal{L}_{\mathcal{U} \to \mathcal{V}} := \{\mathbf{A} : \mathbf{A}\boldsymbol{x} \in \mathcal{V} \text{ for all } \boldsymbol{x} \in \mathcal{U}\}.$$

Finally, we remark that for a matrix $\mathbf{A}$ whose columns are indexed using sequences $\sigma \in \Sigma$, we represent its columns as $\mathbf{A}_{(\sigma)}$. Furthermore, for sequence-form strategies $\boldsymbol{x} \in \mathcal{Q}$, we use $\boldsymbol{x}[\sigma]$ to represent their entries, and $\boldsymbol{x}[\Sigma_{\succeq j}]$ to represent a vector consisting only of the entries corresponding to sequences $\sigma \in \Sigma_{\succeq j}$.

## B.2  A Key Tool: Linear Transformations of Cartesian Products

We are now ready to introduce the following Proposition, which will play an important role in the proof of Theorem 3.1.

**Proposition B.4.** *Let $\mathcal{U}_1, \ldots, \mathcal{U}_m$ be sets, with $\boldsymbol{0} \notin \text{aff }\mathcal{U}_i$ [2] for all $i = 1, \ldots, m$. Furthermore, for any $i = 1, \ldots, m$ and any $\boldsymbol{b}_i \in \mathbb{R}^k$, let $\mathcal{M}_{\mathcal{U}_i \to \mathcal{P}(\boldsymbol{b}_i)}$ be such that $\mathcal{M}_{\mathcal{U}_i \to \mathcal{P}(\boldsymbol{b}_i)} \cong_{\mathcal{U}_i} \mathcal{L}_{\mathcal{U}_i \to \mathcal{P}(\boldsymbol{b}_i)}$. Then, for all $\boldsymbol{b} \in \mathbb{R}^k$,*

$$\mathcal{L}_{(\mathcal{U}_1 \times \cdots \times \mathcal{U}_m) \to \mathcal{P}(\boldsymbol{b})} \cong_{(\mathcal{U}_1 \times \cdots \times \mathcal{U}_m)} \left\{ \begin{bmatrix} \mathbf{A}_1 \mid \cdots \mid \mathbf{A}_m \end{bmatrix} : \begin{matrix} (11) \ \mathbf{A}_i \in \mathcal{M}_{\mathcal{U}_i \to \mathcal{P}(\boldsymbol{b}_i)} & \forall i \in \{1, \ldots, m\} \\ (12) \ \boldsymbol{b}_1 + \cdots + \boldsymbol{b}_m = \boldsymbol{b} & \\ (13) \ \boldsymbol{b}_i \in \mathbb{R}^k & \forall i \in \{1, \ldots, m\} \end{matrix} \right\}.$$
(14)

---

[2] Instead of the condition $\boldsymbol{0} \notin \text{aff }\mathcal{U}_i$, we could equivalently state that there exists $\boldsymbol{\tau}_i$ such that $\boldsymbol{\tau}_i^\top \boldsymbol{x}_i = 1$ for all $\boldsymbol{x}_i \in \mathcal{U}_i$ using the properties of affine sets.

*Proof.* We prove the result by showing the two directions of the inclusion separately.

($\supseteq$) First, we show that for any $b \in \mathbb{R}^k$, any matrix $\mathbf{A} = [\mathbf{A}_1 \mid \cdots \mid \mathbf{A}_m]$ that belongs to the set on the right-hand side of (14) induces a linear transformation from $\mathcal{U}_1 \times \cdots \times \mathcal{U}_m$ to $\mathcal{P}(b)$ and thus belongs to $\mathcal{L}_{(\mathcal{U}_1 \times \cdots \times \mathcal{U}_m) \to \mathcal{P}(b)}$. To that end, we note that for any $x = (x_1, \ldots, x_m) \in \mathcal{U}_1 \times \cdots \times \mathcal{U}_m$,

$$\mathbf{A}x = \sum_{i=1}^m \mathbf{A}_i x_i \overset{(11)}{\geq} 0, \quad \text{and} \quad \mathbf{P}(\mathbf{A}x) = \sum_{i=1}^m \mathbf{PA}_i x_i \overset{(11)}{=} \sum_{i=1}^m b_i \overset{(12)}{=} b,$$

where in both cases we used the fact that $\mathbf{A}_i$ maps any point in $\mathcal{U}_i$ to a point in $\mathcal{P}(b_i) = \{y : \mathbf{P}y = b_i, y \geq 0\}$ by (11). Hence $\mathbf{A}x \in \mathcal{P}(b)$ for all $x \in \mathcal{U}_1 \times \cdots \times \mathcal{U}_m$, as we wanted to show.

($\subseteq$) We now look at the converse, showing that for any $b \in \mathbb{R}^k$ and matrix $\mathbf{B} = [\mathbf{B}_1 \mid \cdots \mid \mathbf{B}_m] \in \mathcal{L}_{(\mathcal{U}_1 \times \cdots \times \mathcal{U}_m) \to \mathcal{P}(b)}$, there exists a matrix $\mathbf{A} = [\mathbf{A}_1 \mid \cdots \mid \mathbf{A}_m]$ that satisfies constraints (11)-(13) and such that $\mathbf{B}x = \mathbf{A}x$ for all $x \in \mathcal{U}_1 \times \cdots \times \mathcal{U}_m$. As a first step, in the next lemma we show that $\mathbf{B}$ is always equivalent to another matrix $\mathbf{B}' = [\mathbf{B}'_1 \mid \cdots \mid \mathbf{B}'_m] \cong_{(\mathcal{U}_1 \times \cdots \times \mathcal{U}_m)} \mathbf{B}$ that satisfies $\mathbf{B}'_i x_i \geq 0$ for all $i = 1, \ldots, m$ and $x_i \in \mathcal{U}_i$.

**Lemma B.5.** *There exist* $\mathbf{B}'_1, \ldots, \mathbf{B}'_n$, *such that* $\mathbf{B} \cong_{(\mathcal{U}_1 \times \cdots \times \mathcal{U}_m)} \mathbf{B}' := [\mathbf{B}'_1 \mid \ldots \mid \mathbf{B}'_n]$, *and furthermore, for all* $i = 1, \ldots, m$, $\mathbf{B}'_i x_i \geq 0$ *for all* $x_i \in \mathcal{U}_i$.

*Proof of Lemma B.5.* Since $0 \notin \text{aff} \, \mathcal{U}_i$ for all $i = 1, \ldots, m$ by hypothesis, then there exist vectors $\tau_i$ such that $\tau_i^\top x_i = 1$ for all $x_i \in \mathcal{U}_i$. For any $k \in \{1, \ldots, d\}$ and $i \in \{1, \ldots, m-1\}$, let

$$\beta_i[k] := \min_{x \in \mathcal{U}_i} (\mathbf{B}_i x)[k] \quad \forall k \in \{1, \ldots, d\}, \qquad \mathbf{B}'_i := \mathbf{B}_i - \beta_i \tau_i^\top$$

Furthermore, let $\beta_m := \sum_{i=1}^{m-1} \beta_i$ and $\mathbf{B}'_m := \mathbf{B}_i + \beta_m \tau_m^\top$. It is immediate to check that the matrix $\mathbf{B}' := [\mathbf{B}'_1 \mid \cdots \mid \mathbf{B}'_m]$ is such that $\mathbf{B}'x = \mathbf{B}x$ for all $x \in \mathcal{U}_1 \times \cdots \times \mathcal{U}_m$, that is, $\mathbf{B}' \cong_{(\mathcal{U}_1 \times \cdots \times \mathcal{U}_m)} \mathbf{B}$. We now show that $\mathbf{B}'_i x_i \geq 0$ for all $i = 1, \ldots, m$ and $x_i \in \mathcal{U}_i$. Expanding the definition of $\mathbf{B}'_i$ and $\beta_i$, for all $i \in \{1, \ldots, m-1\}, x_i \in \mathcal{U}_i$ and $k \in \{1, \ldots, d\}$,

$$(\mathbf{B}'_i x_i)[k] = (\mathbf{B}_i x_i)[k] - (\beta_i)[k] \cdot (\tau_i^\top x_i) = (\mathbf{B}_i x_i)[k] - \min_{\hat{x}_i \in \mathcal{U}_i} (\mathbf{B}_i \hat{x}_i)[k] \geq 0.$$

Hence, it only remains to prove that the same holds for $i = m$. To that end, fix any $k \in \{1, \ldots, d\}$ and $x_m \in \mathcal{U}_m$, and let $x_i^* \in \arg\min_{x_i \in \mathcal{U}_i} (\mathbf{B}_i x_i)[k]$ for all $i \in \{1, \ldots, m-1\}$. Using the fact that all vectors in $\mathcal{P}(b)$ are nonnegative, $x^* := (x_1^*, \ldots, x_{m-1}^*, x_m) \in \mathcal{U}_1 \times \cdots \times \mathcal{U}_m$ must satisfy $\mathbf{B}x^* \geq 0$. Hence,

$$0 \leq (\mathbf{B}x^*)[k] = (\mathbf{B}_m x_m)[k] + \sum_{i=1}^{m-1} (\mathbf{B}_i x_i^*)[k] = (\mathbf{B}_m x_m)[k] + \sum_{i=1}^{m-1} \beta_i[k] = (\mathbf{B}'_m x_m)[k],$$

thus concluding the proof of the lemma. $\qquad\square$

Since $\mathbf{B}' \cong_{(\mathcal{U}_1 \times \cdots \times \mathcal{U}_m)} \mathbf{B}$, and $\mathbf{B}$ maps to $\mathcal{P}(b)$, for all $x = (x_1, \ldots, x_m) \in \mathcal{U}_1 \times \cdots \times \mathcal{U}_m$ we must have

$$b = \mathbf{P}(\mathbf{B}'x) = \sum_{i=1}^m \mathbf{PB}'_i x_i.$$

Since we can pick the $x_i$ for different indices $i$ independently, it follows that $\mathbf{PB}_i x_i$ must be a constant function of $x_i \in \mathcal{U}_i$, that is, there must exist vectors $b_1, \ldots, b_m \in \mathbb{R}^k$ such that

$$b_1 + \cdots + b_m = b, \quad \text{and} \quad \mathbf{PB}'_i x_i = b_i \quad \forall x_i \in \mathcal{U}_i.$$

Since in addition $\mathbf{B}'_i x_i \geq 0$ (by construction of the $\mathbf{B}'_i$), this means that $\mathbf{B}'_i \in \mathcal{L}_{\mathcal{U}_i \to \mathcal{P}(b_i)}$. Finally, using the hypothesis that $\mathcal{L}_{\mathcal{U}_i \to \mathcal{P}(b_i)} \cong_{\mathcal{U}_i} \mathcal{M}_{\mathcal{U}_i \to \mathcal{P}(b_i)}$, there must exist $\mathbf{A}_i \in \mathcal{M}_{\mathcal{U}_i \to \mathcal{P}(b_i)}$, with $\mathbf{A}_i \cong_{\mathcal{U}_i} \mathbf{B}'_i$, for all $i = 1, \ldots, m$. This concludes the proof. $\qquad\square$

## B.3 Characterization of Linear Functions of Subtrees

The following result can be understood as a version of Theorem 3.1 stated for each subtree, rooted at some decision node, of the decision space.

**Theorem B.6.** *For any decision node $j \in \mathcal{J}$ and vector $\boldsymbol{b}_j \in \mathbb{R}^k$, let*

$$
\tilde{\mathcal{M}}_{\mathcal{Q}_{\succeq j} \to \mathcal{P}(\boldsymbol{b}_j)} := \left\{ \underbrace{\left[ \cdots \mid \mathbf{A}_{(ja)} \mid \cdots \right]}_{\in \mathbb{R}^{d \times \Sigma_{\succeq j}}} : 
\begin{array}{lll}
(15) & \mathbf{PA}_{(j'a')} = \boldsymbol{b}_{j'} & \forall j'a' \in \Sigma_{\succeq j} \cap \Sigma^{\perp} \\
(16) & \mathbf{A}_{(j'a')} = \mathbf{0} & \forall j'a' \in \Sigma_{\succeq j} \setminus \Sigma^{\perp} \\
(17) & \sum_{j'' \in \mathcal{C}_{j'a'}} \boldsymbol{b}_{j''} = \boldsymbol{b}_{j'} & \forall j'a' \in \Sigma_{\succeq j} \setminus \Sigma^{\perp} \\
(18) & \mathbf{A}_{(j'a')} \geq \mathbf{0} & \forall j'a' \in \Sigma_{\succeq j} \\
(19) & \boldsymbol{b}_{j'} \in \mathbb{R}^k & \forall j' \succ j
\end{array}
\right\}. \tag{20}
$$

*Then,* $\tilde{\mathcal{M}}_{\mathcal{Q}_{\succeq j} \to \mathcal{P}(\boldsymbol{b}_j)} \cong_{\mathcal{Q}_{\succeq j}} \mathcal{L}_{\mathcal{Q}_{\succeq j} \to \mathcal{P}(\boldsymbol{b}_j)}.$

Before continuing with the proof, we remark a subtle point: unlike (7), which constraints each column to have entries in $[0, \gamma]$, (18) only specifies the lower bound at zero, but no upper bound. Hence the tilde above the symbol of this Theorem. Consequently, the matrices in the set $\tilde{\mathcal{M}}_{\mathcal{Q}_{\succeq j} \to \mathcal{P}(\boldsymbol{b}_j)}$ need not have bounded entries. In that sense, Theorem B.6 is slightly different from Theorem 3.1. We will strengthen (18) to enforce a bound on each column when completing the proof of Theorem 3.1 in the next subsection.

*Proof.* To aid us with the proof, we first express the definition of $\tilde{\mathcal{M}}_{\mathcal{Q}_{\succeq j} \to \mathcal{P}(\boldsymbol{b}_j)}$ in a way that better captures the inductive structure we need. By direct inspection of the constraints, the set $\tilde{\mathcal{M}}_{\mathcal{Q}_{\succeq j} \to \mathcal{P}(\boldsymbol{b}_j)}$ satisfies the inductive definition

$$
\tilde{\mathcal{M}}_{\mathcal{Q}_{\succeq j} \to \mathcal{P}(\boldsymbol{b}_j)} = \left\{
\begin{array}{l}
\mathbf{A} = \left[ \cdots \mid \mathbf{A}_{(ja)} \mid \cdots \right] \in \mathbb{R}^{d \times \Sigma_{\succeq j}}, \text{ such that:} \\[4pt]
\begin{array}{lll}
(21) & \mathbf{PA}_{(ja)} = \boldsymbol{b}_j & \forall a \in \mathcal{A}_j : ja \in \Sigma_{\succeq j} \cap \Sigma^{\perp} \\
(22) & \mathbf{A}_{(ja)} = \mathbf{0} & \forall a \in \mathcal{A}_j : ja \notin \Sigma^{\perp} \\
(23) & \sum_{j' \in \mathcal{C}_{ja}} \boldsymbol{b}_{j'} = \boldsymbol{b}_j & \forall a \in \mathcal{A}_j : ja \notin \Sigma^{\perp} \\
(24) & [\mathbf{A}_{(\sigma)}]_{\sigma \succeq j'} \in \tilde{\mathcal{M}}_{\mathcal{Q}_{\succeq j'} \to \mathcal{P}(\boldsymbol{b}_{j'})} & \forall a \in \mathcal{A}_j, j' \in \mathcal{C}_{ja} \\
(25) & \mathbf{A}_{(ja)} \geq \mathbf{0} & \forall a \in \mathcal{A}_j \\
(26) & \boldsymbol{b}_{j'} \in \mathbb{R}^k & \forall a \in \mathcal{A}_j, j' \in \mathcal{C}_{ja}
\end{array}
\end{array}
\right\}. \tag{27}
$$

We prove the result by structural induction on the tree-form decision process.

- **Base case.** We start by establishing the result for any terminal decision node $j \in \mathcal{J}$, that is, one for which all sequences $\{ja : a \in \mathcal{A}_j\}$ are terminal. In this case, the set $\mathcal{Q}_{\succeq j}$ is the probability simplex $\Delta(\{ja : a \in \mathcal{A}_j\})$. Thus, for a matrix $\mathbf{A}$ to map all $\boldsymbol{x} \in \mathcal{Q}_{\succeq j}$ to elements in the convex polytope $\mathcal{P}(\boldsymbol{b}_j)$ it is both necessary and sufficient that all columns of $\mathbf{A}$ be elements of $\mathcal{P}(\boldsymbol{b}_j)$. It is necessary because if $\mathbf{A}\boldsymbol{x} \in \mathcal{P}(\boldsymbol{b}_j)$ for all $\boldsymbol{x} \in \mathcal{Q}_{\succeq j}$, then for the indicator vector $\boldsymbol{x}$ with $\boldsymbol{x}[ja] = 1$ we get $\mathbf{A}\boldsymbol{x} = \mathbf{A}_{(ja)} \in \mathcal{P}(\boldsymbol{b}_j)$. And, it is sufficient because any $\boldsymbol{x} \in \mathcal{Q}_{\succeq j}$ represents a convex combination of the columns $\mathbf{A}_{(ja)}$.

  The set defined by these constraints matches exactly the set $\tilde{\mathcal{M}}_{\mathcal{Q}_{\succeq j} \to \mathcal{P}(\boldsymbol{b}_j)}$ defined in the statement: since all sequences $ja$ are terminal, in this case it reduces to

  $$
  \tilde{\mathcal{M}}_{\mathcal{Q}_{\succeq j} \to \mathcal{P}(\boldsymbol{b}_j)} = \left\{ \left[ \cdots \mid \mathbf{A}_{(ja)} \mid \cdots \right] \in \mathbb{R}^{d \times \Sigma_{\succeq j}} : 
  \begin{array}{lll}
  (28) & \mathbf{PA}_{(ja)} = \boldsymbol{b}_j & \forall a \in \mathcal{A}_j \\
  (29) & \mathbf{A}_{(ja)} \geq \mathbf{0} & \forall a \in \mathcal{A}_j
  \end{array}
  \right\},
  $$

  that is, the set of matrices whose columns are elements of $\mathcal{P}(\boldsymbol{b}_j)$. So, we have $\tilde{\mathcal{M}}_{\mathcal{Q}_{\succeq j} \to \mathcal{P}(\boldsymbol{b}_j)} = \mathcal{L}_{\mathcal{Q}_{\succeq j} \to \mathcal{P}(\boldsymbol{b}_j)}$ with equality, which immediately implies the claim $\tilde{\mathcal{M}}_{\mathcal{Q}_{\succeq j} \to \mathcal{P}(\boldsymbol{b}_j)} \cong_{\Sigma_{\succeq j}} \mathcal{L}_{\mathcal{Q}_{\succeq j} \to \mathcal{P}(\boldsymbol{b}_j)}.$

- **Inductive step.** We now look at a general decision node $j \in \mathcal{J}$, assuming as inductive hypothesis that the claim holds for any $j' \succ j$. Below we prove that $\tilde{\mathcal{M}}_{\mathcal{Q}_{\succeq j} \to \mathcal{P}(\boldsymbol{b}_j)} \cong_{\mathcal{Q}_{\succeq j}} \mathcal{L}_{\mathcal{Q}_{\succeq j} \to \mathcal{P}(\boldsymbol{b}_j)}$ as well.

($\subseteq$) We start by showing that for any $\boldsymbol{b}_j \in \mathbb{R}^k$, $\boldsymbol{x} \in \Pi_{\succeq j}$ and $\mathbf{A} \in \tilde{\mathcal{M}}_{\mathcal{Q}_{\succeq j} \to \mathcal{P}(\boldsymbol{b}_j)}$, we have $\mathbf{A}\boldsymbol{x} \in \mathcal{P}(\boldsymbol{b}_j)$. From (18) it is immediate that $\mathbf{A}$ has nonnegative entries, and since any vector $\boldsymbol{x} \in \Pi_{\succeq j}$ also has nonnegative entries, it follows that $\mathbf{A}\boldsymbol{x} \geq \mathbf{0}$. Hence, it only remains to show that $\mathbf{P}(\mathbf{A}\boldsymbol{x}) = \boldsymbol{b}_j$. Using Lemma A.2, for any $j' \in \sqcup_{a \in \mathcal{A}_j} \mathcal{C}_{ja}$ there exists $\boldsymbol{x}_{\succeq j'} \in \mathcal{Q}_{\succeq j'}$ such that $\boldsymbol{x}[\Sigma_{\succeq j'}] = \boldsymbol{x}[ja] \cdot \boldsymbol{x}_{\succeq j'}$. Hence, we have

$$
\mathbf{P}(\mathbf{A}\boldsymbol{x}) = \sum_{a \in \mathcal{A}_j} \mathbf{P}\mathbf{A}_{(ja)}\boldsymbol{x}[ja] + \sum_{\substack{a \in \mathcal{A}_j \\ ja \notin \Sigma^{\perp}}} \sum_{j' \in \mathcal{C}_{ja}} \mathbf{P}\mathbf{A}_{\succeq j'}(\boldsymbol{x}[ja] \cdot \boldsymbol{x}_{\succeq j'})
$$

$$
= \sum_{\substack{a \in \mathcal{A}_j \\ ja \in \Sigma^{\perp}}} \boldsymbol{x}[ja] \cdot \mathbf{P}\mathbf{A}_{(ja)} + \sum_{\substack{a \in \mathcal{A}_j \\ ja \notin \Sigma^{\perp}}} \left( \boldsymbol{x}[ja] \sum_{j' \in \mathcal{C}_{ja}} \mathbf{P}\mathbf{A}_{\succeq j'}\boldsymbol{x}_{\succeq j'} \right) \qquad \text{(from (22))}
$$

$$
= \sum_{\substack{a \in \mathcal{A}_j \\ ja \in \Sigma^{\perp}}} \boldsymbol{x}[ja] \cdot \boldsymbol{b}_j + \sum_{\substack{a \in \mathcal{A}_j \\ ja \notin \Sigma^{\perp}}} \left( \boldsymbol{x}[ja] \sum_{j' \in \mathcal{C}_{ja}} \boldsymbol{b}_{j'} \right) \qquad \text{(from (21) and (24))}
$$

$$
= \sum_{\substack{a \in \mathcal{A}_j \\ ja \in \Sigma^{\perp}}} \boldsymbol{x}[ja] \cdot \boldsymbol{b}_j + \sum_{\substack{a \in \mathcal{A}_j \\ ja \notin \Sigma^{\perp}}} \boldsymbol{x}[ja] \cdot \boldsymbol{b}_j \qquad \text{(from (23))}
$$

$$
= \boldsymbol{b}_j \cdot \sum_{a \in \mathcal{A}_j} \boldsymbol{x}[ja] = \boldsymbol{b}_j. \qquad \text{(from Definition 2.2)}
$$

($\supseteq$) Conversely, consider any $\mathbf{B} \in \mathcal{L}_{\mathcal{Q}_{\succeq j} \to \mathcal{P}(\boldsymbol{b}_j)}$. We will show that there exists a matrix $\mathbf{A} \in \tilde{\mathcal{M}}_{\mathcal{Q}_{\succeq j} \to \mathcal{P}(\boldsymbol{b}_j)}$ such that $\mathbf{B} \cong_{\mathcal{Q}_{\succeq j}} \mathbf{A}$. First, we argue that there exists a matrix $\mathbf{B}' \cong_{\mathcal{Q}_{\succeq j}} \mathbf{B}$ with the property that the column $\mathbf{B}'_{(ja)}$ corresponding to any *nonterminal* sequence is identically zero.

**Lemma B.7.** *There exists* $\mathbf{B}' \cong_{\mathcal{Q}_{\succeq j}} \mathbf{B}$ *such that* $\mathbf{B}'_{(ja)} = \mathbf{0}$ *for all* $a \in \mathcal{A}_j$ *such that* $ja \in \Sigma_{\succeq j} \setminus \Sigma^{\perp}$.

*Proof.* Fix any $a \in \mathcal{A}_j$ such that $ja$ is nonterminal. Then, by definition there exists at least one decision node $j'$ whose parent sequence is $ja$. Consider now the matrix $\mathbf{B}''$ obtained from "spreading" column $\mathbf{B}_{(ja)}$ onto $\mathbf{B}_{(j'a')}$ $(a' \in \mathcal{A}_{j'})$, that is, the matrix whose columns are defined according to the following rules: (i) $\mathbf{B}''_{(ja)} = \mathbf{0}$, (ii) $\mathbf{B}''_{(j'a')} = \mathbf{B}_{(j'a')} + \mathbf{B}_{(ja)}$ for all $a' \in \mathcal{A}_{j'}$, (iii) $\mathbf{B}''_{(\sigma)} = \mathbf{B}_{(\sigma)}$ everywhere else. The column $\mathbf{B}''_{(ja)}$ is identically zero by construction, and all other columns $\mathbf{B}''_{(ja')}$, $a' \in \mathcal{A}_j \setminus \{a\}$, are the same as $\mathbf{B}$. Most importantly, since from the sequence-form constraints Definition 2.2 any sequence-form strategy $\boldsymbol{x} \in \mathcal{Q}_{\succeq j}$ satisfies the equality $\boldsymbol{x}[ja] = \sum_{a' \in \mathcal{A}_{j'}} \boldsymbol{x}[j'a']$, the matrix $\mathbf{B}''$ satisfies $\mathbf{B}''\boldsymbol{x} = \mathbf{B}\boldsymbol{x}$ for all $\boldsymbol{x} \in \mathcal{Q}_{\succeq j}$, *i.e.*, $\mathbf{B}'' \cong_{\mathcal{Q}_{\succeq j}} \mathbf{B}$. Iterating the argument for all actions $a \in \mathcal{A}_j$ yields the statement. $\qquad \square$

Consider now any $a \in \mathcal{A}_j$ that leads to a terminal sequence $ja \in \Sigma_{\succeq j} \cap \Sigma^{\perp}$. The vector $\mathbf{1}_{ja}$ defined as having a $1$ in the position corresponding to $ja$ and $0$ everywhere else is a valid sequence-form strategy vector, that is, $\mathbf{1}_{ja} \in \mathcal{Q}_{\succeq j}$. Hence, since $\mathbf{B}'$ maps $\mathcal{Q}_{\succeq j}$ to $\mathcal{P}(\boldsymbol{b}_j)$, it is necessary that $\mathbf{B}'_{(ja)} \in \mathcal{P}(\boldsymbol{b}_j)$, that is, $\mathbf{B}'_{(ja)} \geq \mathbf{0}$ and $\mathbf{P}\mathbf{B}'_{(ja)} = \boldsymbol{b}_j$. In other words, we have just proved the following.

**Lemma B.8.** *For any* $a \in \mathcal{A}_j$ *such that* $ja \in \Sigma_{\succeq j} \cap \Sigma^{\perp}$, $\mathbf{B}'_{(ja)} \geq \mathbf{0}$ *and* $\mathbf{P}\mathbf{B}'_{(ja)} = \boldsymbol{b}_j$.

Combined, Lemmas B.7 and B.8 show that $\mathbf{B}'$ satisfies constraints (21), (22), and (25). Consider now any action $a \in \mathcal{A}_j$ that defines a *nonterminal* sequence $ja \in \Sigma_{\succeq j} \setminus \Sigma^\perp$. For each child decision point $j' \in \mathcal{C}_{ja}$, let $\boldsymbol{x}_{\succeq j'} \in \mathcal{Q}_{\succeq j'}$ be a choice of strategy for that decision point, and denote $\mathbf{B}'_{\succeq j'}$ the submatrix of $\mathbf{B}'$ obtained by only considering the columns $\mathbf{B}'_{(\sigma)}$ corresponding to sequences $\sigma \succeq j'$. The vector $\boldsymbol{x}$ defined according to $\boldsymbol{x}[ja] = 1$, $\boldsymbol{x}[\Sigma_{\succeq j'}] = \boldsymbol{x}_{\succeq j'}$ for all $j' \in \mathcal{C}_{ja}$, and 0 everywhere else is a valid sequence-form strategy $\boldsymbol{x} \in \mathcal{Q}_{\succeq j}$, and therefore $\mathbf{B}'\boldsymbol{x} \in \mathcal{P}(\boldsymbol{b}_j)$ since $\mathbf{B}' \cong_{\mathcal{Q}_{\succeq j}} \mathbf{B}$ and $\mathbf{B} \in \mathcal{L}_{\mathcal{Q}_{\succeq j} \to \mathcal{P}(\boldsymbol{b}_j)}$ by hypothesis. Therefore, using the fact that $\mathbf{B}'_{(ja)} = \mathbf{0}$ by Lemma B.7, we conclude that

$$\mathcal{P}(\boldsymbol{b}_j) \ni \mathbf{B}'\boldsymbol{x} = \sum_{j' \in \mathcal{C}_{ja}} \mathbf{B}'_{\succeq j'}\boldsymbol{x}_{\succeq j'}.$$

Because the above holds for any choice of $\boldsymbol{x}_{\succeq j'} \in \mathcal{Q}_{\succeq j'}$, it follows that the matrix $[\cdots \mid \mathbf{B}'_{\succeq j'} \mid \cdots] \in \mathcal{L}_{\times_{j' \in \mathcal{C}_{ja}} \mathcal{Q}_{\succeq j'} \to \mathcal{P}(\boldsymbol{b}_j)}$. Hence, applying Proposition B.4 (note that $\mathbf{0} \notin \text{aff } \mathcal{Q}_{\succeq j'}$ since $\sum_{a \in \mathcal{A}_{j'}} \boldsymbol{x}[j'a] = 1$ for all $\boldsymbol{x} \in \mathcal{Q}_{\succeq j'}$ by Definition A.1) together with the inductive hypothesis, we conclude that for each $j' \in \mathcal{C}_{ja}$ there exist a vector $\boldsymbol{b}_{j'} \in \mathbb{R}^k$ and a matrix $\mathbf{A}_{\succeq j'} \in \mathcal{M}_{\mathcal{Q}_{\succeq j'} \to \mathcal{P}(\boldsymbol{b}_{j'})}$, such that $\sum_{j' \in \mathcal{C}_{ja}} \boldsymbol{b}_{j'} = \boldsymbol{b}_j$ and $[\cdots \mid \mathbf{B}'_{\succeq j'} \mid \cdots] \cong_{\times_{j' \in \mathcal{C}_{ja}} \mathcal{Q}_{\succeq j'}} [\cdots \mid \mathbf{A}_{\succeq j'} \mid \cdots]$. We can therefore replace all columns corresponding to $\mathbf{B}'_{\succeq j'}$ with those of $\mathbf{A}_{\succeq j'}$, obtaining a new matrix $\cong_{\mathcal{Q}_{\succeq j}} \mathbf{B}'$. Repeating the argument for each $ja \in \Sigma_{\succeq j} \setminus \Sigma^\perp$ finally yields a new matrix that is $\cong_{\Sigma_{\succeq j}} \mathbf{B}$ and satisfies all constraints given in (27), as we wanted to show. $\qquad\square$

## B.4 Putting all the Pieces Together

Finally, we are ready to prove the main result of the paper.

**Theorem 3.1.** *Let $\mathcal{Q}$ be a sequence-form strategy space and let $\mathcal{P}$ be any bounded polytope of the form $\mathcal{P} := \{\boldsymbol{x} \in \mathbb{R}^d : \mathbf{P}\boldsymbol{x} = \boldsymbol{p}, \boldsymbol{x} \geq \mathbf{0}\} \subseteq [0, \gamma]^d$, where $\mathbf{P} \in \mathbb{R}^{k \times d}$. Then, for any linear function $f : \mathcal{Q} \to \mathcal{P}$, there exists a matrix $\mathbf{A}$ in the polytope*

$$\mathcal{M}_{\mathcal{Q} \to \mathcal{P}} := \left\{ \mathbf{A} = [\cdots \mid \mathbf{A}_{(\sigma)} \mid \cdots] \in \mathbb{R}^{d \times \Sigma} : \begin{array}{lll} (3) & \mathbf{PA}_{(ja)} = \boldsymbol{b}_j & \forall\, ja \in \Sigma^\perp \\ (4) & \mathbf{A}_{(\sigma)} = \mathbf{0} & \forall\, \sigma \in \Sigma \setminus \Sigma^\perp \\ (5) & \sum_{j' \in \mathcal{C}_\varnothing} \boldsymbol{b}_{j'} = \boldsymbol{p} & \\ (6) & \sum_{j' \in \mathcal{C}_{ja}} \boldsymbol{b}_{j'} = \boldsymbol{b}_j & \forall\, ja \in \Sigma \setminus \Sigma^\perp \\ (7) & \mathbf{A}_{(\sigma)} \in [0, \gamma]^d & \forall\, \sigma \in \Sigma \\ (8) & \boldsymbol{b}_j \in \mathbb{R}^k & \forall\, j \in \mathcal{J} \end{array} \right\}$$

*such that $f(\boldsymbol{x}) = \mathbf{A}\boldsymbol{x}$ for all $\boldsymbol{x} \in \mathcal{Q}$. Conversely, any $\mathbf{A} \in \mathcal{M}_{\mathcal{Q} \to \mathcal{P}}$ defines a linear function $\boldsymbol{x} \mapsto \mathbf{A}\boldsymbol{x}$ from $\mathcal{Q}$ to $\mathcal{P}$, that is, such that $\mathbf{A}\boldsymbol{x} \in \mathcal{P}$ for all $\boldsymbol{x} \in \mathcal{Q}$.*

*Proof of Theorem 3.1.* We prove the result in two steps. First, we show that

$$\mathcal{L}_{\mathcal{Q} \to \mathcal{P}} \cong_{\mathcal{Q}} \tilde{\mathcal{M}}_{\mathcal{Q} \to \mathcal{P}} := \left\{ \mathbf{A} = [\cdots \mid \mathbf{A}_{(\sigma)} \mid \cdots] \in \mathbb{R}^{d \times \Sigma} : \begin{array}{lll} (30) & \mathbf{PA}_{(ja)} = \boldsymbol{b}_j & \forall\, ja \in \Sigma^\perp \\ (31) & \mathbf{A}_{(\sigma)} = \mathbf{0} & \forall\, \sigma \in \Sigma \setminus \Sigma^\perp \\ (32) & \sum_{j' \in \mathcal{C}_\varnothing} \boldsymbol{b}_{j'} = \boldsymbol{p} & \\ (33) & \sum_{j' \in \mathcal{C}_{ja}} \boldsymbol{b}_{j'} = \boldsymbol{b}_j & \forall\, ja \in \Sigma \setminus \Sigma^\perp \\ (34) & \mathbf{A}_{(\sigma)} \geq \mathbf{0} & \forall\, \sigma \in \Sigma \\ (35) & \boldsymbol{b}_j \in \mathbb{R}^k & \forall\, j \in \mathcal{J} \end{array} \right\},$$

where the difference between $\tilde{\mathcal{M}}_{\mathcal{Q} \to \mathcal{P}}$ and $\mathcal{M}_{\mathcal{Q} \to \mathcal{P}}$ lies in constraint (34), which only sets a lower bound (at zero) for each entry of the matrix, as opposed to a bound $[0, \gamma]$ as in (7). Using the definition of $\tilde{\mathcal{M}}_{\mathcal{Q}_{\succeq j} \to \mathcal{P}(\boldsymbol{b}_j)}$ given in (20) (Theorem B.6), the set $\tilde{\mathcal{M}}_{\mathcal{Q} \to \mathcal{P}}$ can be equivalently written as

$$\tilde{\mathcal{M}}_{\mathcal{Q} \to \mathcal{P}} = \left\{ \mathbf{A} = [\cdots \mid \mathbf{A}_{(\sigma)} \mid \cdots] \in \mathbb{R}^{d \times \Sigma} : \begin{array}{lll} (36) & \mathbf{A}_{(\varnothing)} = \mathbf{0} & \\ (37) & \sum_{j \in \mathcal{C}_\varnothing} \boldsymbol{b}_j = \boldsymbol{p} & \\ (38) & [\mathbf{A}_{(\sigma)}]_{\sigma \succeq j} \in \tilde{\mathcal{M}}_{\mathcal{Q}_{\succeq j} \to \mathcal{P}(\boldsymbol{b}_j)} & \forall\, j \in \mathcal{C}_\varnothing \\ (39) & \boldsymbol{b}_j \in \mathbb{R}^k & \forall\, j \in \mathcal{C}_\varnothing \end{array} \right\}.$$

To show that $\tilde{\mathcal{M}}_{\mathcal{Q}\to\mathcal{P}} \cong_{\mathcal{Q}} \mathcal{L}_{\mathcal{Q}\to\mathcal{P}}$, we proceed exactly like in the inductive step of the proof of Theorem B.6. Specifically, let $\mathbf{A} \in \tilde{\mathcal{M}}_{\mathcal{Q}\to\mathcal{P}}$ and $\boldsymbol{x} \in \mathcal{Q}$ be arbitrary. From Definition 2.2 it follows that $\boldsymbol{x}[\Sigma_{\succeq j}] \in \mathcal{Q}_{\succeq j}$ for any $j \in \mathcal{C}_\varnothing$ and therefore, denoting $\mathbf{A}_{\succeq j} := [\mathbf{A}_{(\sigma)}]_{\sigma \succeq j}$,

$$\mathbf{P}(\mathbf{A}\boldsymbol{x}) \overset{(36)}{=} \sum_{j\in\mathcal{C}_\varnothing} \mathbf{PA}_{(\sigma)}\boldsymbol{x}[\Sigma_{\succeq j}] \overset{(38)}{=} \sum_{j\in\mathcal{C}_\varnothing} \boldsymbol{b}_j \overset{(38)}{=} \boldsymbol{p}, \qquad \mathbf{A}\boldsymbol{x} \overset{(36)}{=} \sum_{j\in\mathcal{C}_\varnothing} \mathbf{A}_{(\sigma)}\boldsymbol{x}[\Sigma_{\succeq j}] \overset{(38)}{\geq} \mathbf{0}.$$

which shows that $\mathbf{A}\boldsymbol{x} \in \mathcal{P}$. Since $\mathbf{A}$ and $\boldsymbol{x}$ were arbitrary, it follows that $\tilde{\mathcal{M}}_{\mathcal{Q}\to\mathcal{P}} \subseteq \mathcal{L}_{\mathcal{Q}\to\mathcal{P}}$. Conversely, let $\mathbf{B} \in \mathcal{L}_{\mathcal{Q}\to\mathcal{P}}$ be arbitrary, and fix a root decision node $j \in \mathcal{C}_\varnothing$. Then, we can "spread out" the column $\mathbf{B}_{(\varnothing)}$ by adding it to each $\mathbf{B}_{(ja)} : a \in \mathcal{A}_j$ by constructing the matrix $\mathcal{L}_{\mathcal{Q}\to\mathcal{P}} \ni \mathbf{B}' \cong_{\mathcal{Q}} \mathbf{B}$ defined by (i) $\mathbf{B}'_{(\varnothing)} = \mathbf{0}$, (ii) $\mathbf{B}'_{(ja)} = \mathbf{B}_{(ja)} + \mathbf{B}_{(\varnothing)}$ for any $a \in \mathcal{A}_j$, and (iii) $\mathbf{B}'_{(\sigma)} = \mathbf{B}_{(\sigma)}$ everywhere else. Pick now any vectors $\{\boldsymbol{x}_{\succeq j} \in \mathcal{Q}_{\succeq j} : j \in \mathcal{C}_\varnothing\}$, and consider the vector $\boldsymbol{x}$ defined as $\boldsymbol{x}[\varnothing] = 1$, and $\boldsymbol{x}[\Sigma_{\succeq j}] = \boldsymbol{x}_{\succeq j}$ for all $j \in \mathcal{C}_\varnothing$. The vector $\boldsymbol{x}$ is a valid sequence-form strategy, that is, $\boldsymbol{x} \in \mathcal{Q}$. Let now $\mathbf{B}'_{\succeq j} := [\mathbf{B}'_{(\sigma)}]_{\sigma \succeq j}$. From the fact that $\mathbf{B}'\boldsymbol{x} \in \mathcal{P}$, together with the fact that by construction $\mathbf{B}'_{(\varnothing)} = \mathbf{0}$, we conclude that

$$\mathbf{B}'\boldsymbol{x} = \sum_{j\in\mathcal{C}_\varnothing} \mathbf{B}'_{\succeq j}\boldsymbol{x}_{\succeq j} \in \mathcal{P}.$$

Since the inclusion above holds for any choice of $\{\boldsymbol{x}_{\succeq j} \in \mathcal{Q}_{\succeq j} : j \in \mathcal{C}_\varnothing\}$, and since for all $j \in \mathcal{C}_\varnothing$ the vector $\mathbf{0} \notin \text{aff } \mathcal{Q}_{\succeq j}$ (indeed, $\sum_{a\in\mathcal{A}_j} \boldsymbol{x}[ja] = 1$ for all $\boldsymbol{x} \in \Sigma_{\succeq j}$ by Definition A.1), from Proposition B.4 together with Theorem B.6 we conclude that for each $j \in \mathcal{C}_\varnothing$ there exists a vector $\boldsymbol{b}_j \in \mathbb{R}^k$ and a matrix $\mathbf{A}_{\succeq j} \in \tilde{\mathcal{M}}_{\mathcal{Q}_{\succeq j}\to\mathcal{P}(\boldsymbol{b}_j)}$, such that $\sum_{j\in\mathcal{C}_\varnothing} \boldsymbol{b}_j = \boldsymbol{p}$ and $\mathbf{A}_{\succeq j} \cong_{\mathcal{Q}_{\succeq j}} \mathbf{B}'_{\succeq j}$. By replacing the submatrices $\mathbf{B}'_{\succeq j}$ with $\mathbf{A}_{\succeq j}$ in $\mathbf{B}'$ we then obtain an equivalent matrix that satisfies all constraints that define $\tilde{\mathcal{M}}_{\mathcal{Q}\to\mathcal{P}}$. In summary, we have $\tilde{\mathcal{M}}_{\mathcal{Q}\to\mathcal{P}} \cong \mathcal{L}_{\mathcal{Q}\to\mathcal{P}}$.

To conclude the proof, we now show that $\tilde{\mathcal{M}}_{\mathcal{Q}\to\mathcal{P}} = \mathcal{M}_{\mathcal{Q}\to\mathcal{P}}$. First, we make the straightforward observation that any $\mathbf{A} \in \mathcal{M}_{\mathcal{Q}\to\mathcal{P}}$ also belongs to $\tilde{\mathcal{M}}_{\mathcal{Q}\to\mathcal{P}}$, as the constraint that define the latter set are only looser. Hence, we only need to show that any $\mathbf{B} \in \tilde{\mathcal{M}}_{\mathcal{Q}\to\mathcal{P}}$ also satisfies constraint (7). Since $\mathbf{B} \in \tilde{\mathcal{M}}_{\mathcal{Q}\to\mathcal{P}}$, all columns of $\mathbf{B}$ are nonnegative (constraint (34)). Furthermore, since $\tilde{\mathcal{M}}_{\mathcal{Q}\to\mathcal{P}} \cong_{\mathcal{Q}} \mathcal{L}_{\mathcal{Q}\to\mathcal{P}}$, clearly $\mathbf{B}\boldsymbol{x} \in \mathcal{P}$ for all $\boldsymbol{x} \in \mathcal{Q}$. Fix now any sequence $\sigma \in \Sigma$, and consider any strategy $\boldsymbol{x} \in \mathcal{Q}$ that puts proability mass 1 on all the actions on the path from the root to $\sigma$ included, that is, any $\boldsymbol{x} \in \mathcal{Q}$ with $\boldsymbol{x}[\sigma] = 1$. Then, from the nonnegativity of the columns of $\mathbf{B}$, it follows that

$$\mathcal{P} \ni \mathbf{B}\boldsymbol{x} \geq \mathbf{B}_{(\sigma)}.$$

Since by definition of $\gamma$ any point in $\mathcal{P}$ belongs to $[0,\gamma]^d$, we then conclude that $\mathbf{B}_{(\sigma)} \in [0,\gamma]^d$, implying that $\mathbf{B} \in \mathcal{M}_{\mathcal{Q}\to\mathcal{P}}$ as we wanted to show. $\qquad\square$

## C  Corollaries of the Characterization Theorem

We mention two direct corollaries of Theorem 3.1 that slightly extend the scope of the characterization. The first corollary asserts that the polytope $\mathcal{M}_{\mathcal{Q}\to\mathcal{P}}$ characterizes not only all *linear* functions from $\mathcal{Q}$ to $\mathcal{P}$, but also all *affine* functions.

**Corollary C.1** (From linear to affine functions). *Let $\mathcal{Q}$ be a sequence-form strategy space and let $\mathcal{P}$ be any polytope. Then, for any* affine *function $g : \mathcal{Q} \to \mathcal{P}$, there exists a matrix $\mathbf{A}$ in the polytope $\mathcal{M}_{\mathcal{Q}\to\mathcal{P}}$ defined in Theorem 3.1 such that $g(\boldsymbol{x}) = \mathbf{A}\boldsymbol{x}$ for all $\boldsymbol{x} \in \mathcal{Q}$. Conversely, any $\mathbf{A} \in \mathcal{M}_{\mathcal{Q}\to\mathcal{P}}$ induces an affine function from $\mathcal{Q}$ to $\mathcal{P}$.*

*Proof.* The second part of the statement is trivial since any linear function is also affine, and any $\mathbf{A} \in \mathcal{M}_{\mathcal{Q}\to\mathcal{P}}$ induces a linear function from $\mathcal{Q}$ to $\mathcal{P}$. Let $g(\boldsymbol{x}) = f(\boldsymbol{x}) + \boldsymbol{b}$ be any affine function, where $f$ is an appropriate linear function from $\mathcal{Q}$ to $\mathcal{P}$ and $\boldsymbol{b} \in \mathbb{R}^n$. Since $\boldsymbol{q}[\varnothing] = 1$ for all $\boldsymbol{q} \in \mathcal{Q}$, the function $g$ coincides on $\mathcal{Q}$ with the function $\tilde{g} : \mathcal{Q} \ni \boldsymbol{x} \mapsto f(\boldsymbol{x}) + \boldsymbol{b} \cdot \boldsymbol{x}[\varnothing]$, which is a linear function of $\boldsymbol{x}$. Hence, from the first part of Theorem 3.1 there exists $\mathbf{A} \in \mathcal{M}_{\mathcal{Q}\to\mathcal{P}}$ such that $\mathbf{A}\boldsymbol{x} = \tilde{g}(\boldsymbol{x}) = g(\boldsymbol{x})$ for all $\boldsymbol{x} \in \mathcal{Q}$. $\qquad\square$

The second corollary of Theorem 3.1 extends the characterization to the alternative definition of polytope as a bounded set of the form $\mathcal{C} := \{\boldsymbol{y} \in \mathbb{R}^n : \mathbf{C}\boldsymbol{y} \leq \boldsymbol{c}\}$, by first introducing slack variables and rewriting the polytope in the form handled by Theorem 3.1.

**Corollary C.2** (Alternative polytope representations). *Let $Q$ be a sequence-form strategy polytope, and $C := \{y \in \mathbb{R}^n : Cy \le c\} \subseteq [-\gamma, \gamma]^n$ be a bounded polytope, where $C \in \mathbb{R}^{m \times n}$. Let $k := \max\{\|C\|_\infty, \|c\|_\infty\}$ and introduce the polytope*

$$\tilde{\mathcal{P}} := \left\{ (\tilde{y}, s) \in \mathbb{R}^n \times \mathbb{R}^m : \begin{bmatrix} C & | & kn\mathbf{I} \end{bmatrix} \begin{bmatrix} \tilde{y} \\ s \end{bmatrix} = c + \gamma C\mathbf{1}, \quad \begin{bmatrix} \tilde{y} \\ s \end{bmatrix} \ge \mathbf{0} \right\} \subseteq [0, 2\gamma]^{n+m},$$

*which is of the form handled by Theorem 3.1. For any affine function $g : Q \to C$, there exists a matrix $A$ in the polytope*

$$\tilde{\mathcal{M}}_{Q \to C} := \left\{ \begin{bmatrix} \tilde{M}_{(\varnothing)} - \gamma \mathbf{1} & | & \cdots & | & \tilde{M}_{(\sigma)} & | & \cdots \end{bmatrix} \in \mathbb{R}^{n \times \Sigma} : \begin{bmatrix} \tilde{M} \\ \tilde{Z} \end{bmatrix} \in \mathcal{M}_{Q \to \tilde{\mathcal{P}}}, \ \tilde{M} \in \mathbb{R}^{n \times \Sigma}, \ \tilde{Z} \in \mathbb{R}^{m \times \Sigma} \right\}$$

*such that $g(x) = Ax$ for all $x \in Q$. Conversely, any $A \in \tilde{\mathcal{M}}_{Q \to C}$ induces an affine function from $Q$ to $C$.*

*Proof.* We begin by proving that $C \subseteq [-\gamma, \gamma]^n$ implies $\tilde{\mathcal{P}} \subseteq [0, 2\gamma]^{n+m}$. Consider any $(\tilde{y}, s) \in \tilde{\mathcal{P}}$ and set $y = \tilde{y} - \gamma \mathbf{1}$. Then, this is a valid $y \in C \subseteq [-\gamma, \gamma]^n$ and, consequently, $\tilde{y} \in [0, 2\gamma]^n$. For the slack variables $s$ it holds that $kns = c - Cy$, where $y = \tilde{y} - \gamma \mathbf{1}$ from before. By definition of $k$ and by $y \ge -\gamma \mathbf{1}$, we conclude that $c - Cy \le (k + n\gamma k)\mathbf{1} \implies s \in [0, 2\gamma]^m$.

Now let $g : Q \to C$ be any affine function from $Q$ to $C$. Then we can define an affine function $f : Q \to \tilde{\mathcal{P}}$ such that

$$f(x) = \begin{bmatrix} g(x) + \gamma \mathbf{1} \\ s \end{bmatrix}$$

for all $x \in Q$. By Corollary C.1 we know that $\mathcal{M}_{Q \to \tilde{\mathcal{P}}}$ characterizes all affine functions from $Q$ to $\tilde{\mathcal{P}}$, including the previous function $f$. Thus, there exists an $\tilde{M} \in \mathbb{R}^{n \times \Sigma}$ such that $g(x) + \gamma \mathbf{1} = \tilde{M}x$ for all $x \in Q$. Since $x[\varnothing] = 1$ for all $x \in Q$, we conclude that there exists $A \in \tilde{\mathcal{M}}_{Q \to C}$ such that $g(x) = Ax = \tilde{M}x - \gamma \mathbf{1}$ for all $x \in Q$.

Conversely, consider any $A \in \tilde{\mathcal{M}}_{Q \to C}$ and define $g(x) = Ax$. That is, there exist suitable $\tilde{M} \in \mathbb{R}^{n \times \Sigma}, \tilde{Z} \in \mathbb{R}^{m \times \Sigma}$ that satisfy the constraints of polytope $\tilde{\mathcal{M}}_{Q \to C}$. Then for all $x \in Q$ it holds $g(x) = \tilde{M}x - \gamma \mathbf{1}$, and $(\tilde{y}, s) \in \tilde{\mathcal{P}}$ where $\tilde{y} = \tilde{M}x$ and $s = \tilde{Z}x$. Thus, by construction of $\tilde{\mathcal{P}}$, as we also argued in the beginning of the proof, we conclude that $g(x) = \tilde{y} - \gamma \mathbf{1} = y \in C$. $\square$

# D  Additional proofs

**Theorem D.1.** *Let $\Sigma$ denote the set of sequences of the learning player in the extensive-form game, and let $\eta^{(t)} = 1/\sqrt{t}$ for all $t$. Then, for any sequence of loss vectors $\ell^{(t)} \in [0, 1]^\Sigma$, Algorithm 1 guarantees linear-swap regret $O(|\Sigma|^2 \sqrt{T})$ after any number $T$ of iterations, and runs in $O(|\Sigma|^{10} \log(|\Sigma|) \log^2 t)$ time for each iteration $t$.*

*Proof of Theorem 3.2.* First we focus on the linear-swap regret bound. Based on Gordon et al. [2008] the $\Phi$-regret equals external regret over the set $\Phi$ of transformations. In our case $\Phi$ is the set $\mathcal{M}_{Q \to Q}$ of all valid linear transformations and the losses for the external regret minimizer are functions $A \mapsto \langle \ell^{(1)}, Ax^{(1)} \rangle, A \mapsto \langle \ell^{(2)}, Ax^{(2)} \rangle, \ldots$ Equivalently, we can write these as $A \mapsto \langle L^{(t)}, A \rangle_F$, where $\langle \cdot, \cdot \rangle_F$ is the component-wise inner product for matrices and $L^{(t)} = \ell^{(t)}(x^{(t)})^\top$, which is a rank-one matrix. Let $D$ be an upper bound on the diameter of $\mathcal{M}_{Q \to Q}$, and $L$ be such that $\|L^{(t)}\|_F \le L$ for all $t$. Then, based on Orabona [2022] we can bound the external regret for this instance of Online Linear Optimization by picking $\eta^{(t)} = \frac{D}{L\sqrt{t}}$ which gives a regret of $O(DL\sqrt{T})$. Since $A \in [0, 1]^{\Sigma \times \Sigma}$ we get $D = |\Sigma|$, and since $\ell^{(t)}, x^{(t)} \in [0, 1]^\Sigma$ we get $L = |\Sigma|$. This results in the desired linear-swap regret of $O(|\Sigma|^2 \sqrt{T})$.

However, in our previous analysis we assumed that it is possible to compute an exact solution $A^{(t+1)} = \Pi_{\mathcal{M}_{Q \to Q}}(A^{(t)} - \eta^{(t)} L^{(t)})$ of the projection step, which is an instance of convex quadratic programming, meaning that we can only get an approximate solution [Vishnoi, 2021]. In the following Lemma we prove that an $\epsilon$-approximate projection does not affect the regret for a single iteration of

the algorithm, if $\epsilon$ is sufficiently small. The inequality we prove is similar to the one in Lemma 2.12 from Orabona [2022].

**Lemma D.2.** *Let* $\mathbf{Y}^* = \Pi_{\mathcal{M}_{\mathcal{Q}\to\mathcal{Q}}}(\mathbf{A}^{(t)} - \eta^{(t)}\mathbf{L}^{(t)})$ *and suppose that* $\mathbf{A}^{(t+1)} \in \mathcal{M}_{\mathcal{Q}\to\mathcal{Q}}$ *is such that* $\|\mathbf{A}^{(t+1)} - \mathbf{Y}^*\|_F^2 \leq \epsilon^{(t)}$, *then for any* $\mathbf{X} \in \mathcal{M}_{\mathcal{Q}\to\mathcal{Q}}$ *it holds*

$$\langle \mathbf{L}^{(t)}, \mathbf{X} - \mathbf{A}^{(t)} \rangle_F \leq \frac{1}{2\eta^{(t)}}\|\mathbf{A}^{(t)} - \mathbf{X}\|_F^2 - \frac{1}{2\eta^{(t)}}\|\mathbf{A}^{(t+1)} - \mathbf{X}\|_F^2 + \frac{\eta^{(t)}}{2}\|\mathbf{L}^{(t)}\|_F^2 + \frac{D}{\eta^{(t)}}\frac{1}{\epsilon^{(t)}}$$

*Proof.* From Lemma 2.12 of Orabona [2022] we know that

$$\langle \mathbf{L}^{(t)}, \mathbf{X} - \mathbf{A}^{(t)} \rangle_F \leq \frac{1}{2\eta^{(t)}}\|\mathbf{A}^{(t)} - \mathbf{X}\|_F^2 - \frac{1}{2\eta^{(t)}}\|\mathbf{Y}^* - \mathbf{X}\|_F^2 + \frac{\eta^{(t)}}{2}\|\mathbf{L}^{(t)}\|_F^2.$$

Additionally, it holds that

$$\begin{aligned}
\|\mathbf{A}^{(t+1)} - \mathbf{X}\|_F^2 &= \|\mathbf{A}^{(t+1)} - \mathbf{Y}^* + \mathbf{Y}^* - \mathbf{X}\|_F^2 \\
&= \|\mathbf{A}^{(t+1)} - \mathbf{Y}^*\|_F^2 + \|\mathbf{Y}^* - \mathbf{X}\|_F^2 + 2\langle \mathbf{A}^{(t+1)} - \mathbf{Y}^*, \mathbf{Y}^* - \mathbf{X} \rangle_F \\
&\leq \|\mathbf{Y}^* - \mathbf{X}\|_F^2 + 2\langle \mathbf{A}^{(t+1)} - \mathbf{Y}^*, \mathbf{A}^{(t+1)} - \mathbf{Y}^* + \mathbf{Y}^* - \mathbf{X} \rangle_F \\
&= \|\mathbf{Y}^* - \mathbf{X}\|_F^2 + 2\langle \mathbf{A}^{(t+1)} - \mathbf{Y}^*, \mathbf{A}^{(t+1)} - \mathbf{X} \rangle_F \\
&\leq \|\mathbf{Y}^* - \mathbf{X}\|_F^2 + 2\|\mathbf{A}^{(t+1)} - \mathbf{Y}^*\|_F\|\mathbf{A}^{(t+1)} - \mathbf{X}\|_F \\
&\leq \|\mathbf{Y}^* - \mathbf{X}\|_F^2 + 2D\epsilon^{(t)}.
\end{aligned}$$

The Lemma follows by combining the previous two inequalities. $\qquad\square$

If we set $\epsilon^{(t)} = 1/t^{5/2}$ then for our choice of $\eta^{(t)} = \frac{D}{L\sqrt{t}}$, the error term becomes $L/t^2$. Summing over $T$ timesteps we thus get an additive error of $O(L) = O(|\Sigma|)$ in the regret, and our total linear-swap regret bound remains $O(|\Sigma|^2\sqrt{T})$.

We now move to analyzing the per-iteration time complexity of Algorithm 1. The most computationally heavy steps are (5) and (6). To compute the fixed point at step (6) we can use any polynomial-time LP algorithm. We can also perform the projection step (5) in polynomial time using the ellipsoid method [Vishnoi, 2021]. For this we reduce it to a suitable Semidefinite Program with $O(|\Sigma|^2)$ variables and use the Cholesky factorization [Gärtner and Matousek, 2014] as the separation oracle, thus responding to separation queries in $O(|\Sigma|^6)$ time. Note that it is possible to guarantee $\|\mathbf{A}^{(t+1)} - \mathbf{Y}^*\|_F^2 \leq \epsilon^{(t)}$ as the ellipsoid method can output a point $\mathbf{A}^{(t+1)} \in \mathcal{M}_{\mathcal{Q}\to\mathcal{Q}}$ such that $\|\mathbf{A}^{(t)} - \eta^{(t)}\mathbf{L}^{(t)} - \mathbf{A}^{(t+1)}\|_F \leq \|\mathbf{A}^{(t)} - \eta^{(t)}\mathbf{L}^{(t)} - \mathbf{Y}^*\|_F + \epsilon$ and furthermore, the Frobenius norm is a 2-strongly convex function.

To apply the ellipsoid method we further need bounds on the Frobenius norm of $\mathbf{Y} \in \mathcal{M}_{\mathcal{Q}\to\mathcal{Q}}$ and the maximum projection distance. For the norm of $\mathbf{Y}$, we pick an upper bound of $R = D = |\Sigma|$. For the lower bound $r$ we note that $\mathbf{Y}\boldsymbol{x} \in \mathcal{Q}$ for all $\boldsymbol{x} \in \mathcal{Q}$, which implies $\mathbf{Y}[\varnothing]\boldsymbol{x} = 1 \implies \|\mathbf{Y}[\varnothing]\|_1 \geq 1$. Thus we get $r \geq \frac{1}{|\Sigma|}$. Similarly, we bound the projection distance between $0$ and $D$. Based on these bounds and on Theorem 13.1 from Vishnoi [2021] we conclude that the total per-iteration time complexity of Algorithm 1 is $O\left(|\Sigma|^{10}\log(|\Sigma|)\log^2(t)\right)$. $\qquad\square$

**Theorem 4.1.** *For two-player, perfect-recall extensive-form games with chance moves, the problem MAXPAY-LCE is not solvable in polynomial time, unless P=NP.*

*Proof.* We use the exact same argument employed in the paper by von Stengel and Forges [2008] by reducing SAT to the MAXPAY-LCE problem. Specifically, for each instance of SAT having $n$ clauses and $m$ variables we construct a two-player extensive-form game of size polynomial in $n$ and $m$. In the beginning, there is a chance move that picks one of $n$ possible actions uniformly at random – one for each SAT clause. Then is the turn of Player 2 who has $n$ distinct singleton information sets corresponding to the chance node actions, and respectively to the $n$ clauses of the SAT instance. Let $L_i$ be the set of literals (negated or non-negated variables) included in the $i$-th clause of the SAT

instance. In information set $i$ of Player 2 there exist $|L_i|$ actions, one for each literal in $L_i$. Finally, each literal leads to a different decision node for Player 1 who has as many decision nodes as the number of literals in the SAT instance, and in each node there exist exactly 2 possible actions: TRUE and FALSE. However, Player 1 only has $m$ information sets corresponding to the $m$ SAT variables with each information set $x$ grouping together all nodes corresponding to literals of the variable $x$. This way, Player 1 only chooses the truth value of a variable without knowing from which literal Player 2 has picked this variable. The utilities for both players are equal to 1 if the truth value picked by Player 1 satisfies the literal picked by Player 2, and both utilities are 0 otherwise.

In this game, there exists a pure strategy attaining payoff 1 for each player if and only if the SAT instance is satisfiable – namely Player 1 always acts based on the satisfying assignment and Player 2 picks for every clause a literal that is known to be TRUE. Otherwise, the maximum payoff for each pure strategy is at most $1 - 1/n$. Given that a LCE describes a convex combination of pure strategies, it follows that the maximum total expected payoff of the the sum of the two players' utilities will either be 2 when the SAT instance is satisfiable, or it will be at most $2(1 - 1/n)$ when the instance is not satisfiable. Additionally, this holds not just for the sum but for any linear combination of player utilities. Thus, the problem of deciding whether MAXPAY-LCE can attain a value of at least $k$ is NP-hard for any linear combination of utilities. $\qquad\square$

# E    Examples

**Example E.1** (EFCE $\neq$ LCE). *Consider the following 2-player signaling game presented by von Stengel and Forges [2008]*

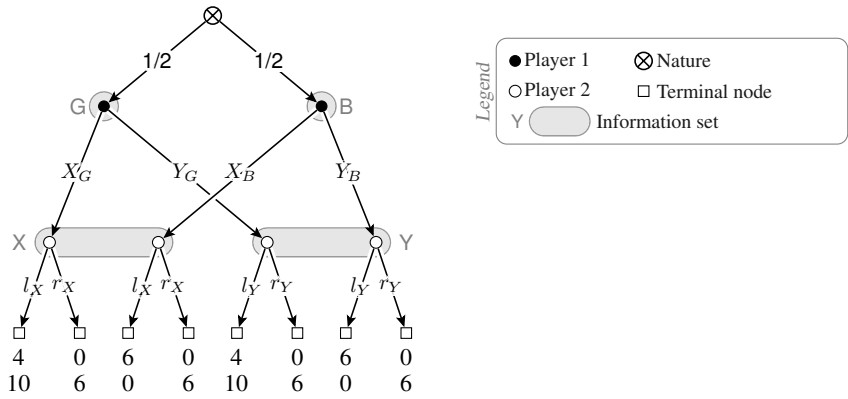

*Below are the normal-form payoff matrix of the signaling game (left) and the extensive-form correlated equilibrium (right) given by von Stengel and Forges [2008]:*

| 1 \ 2 | $l_X l_Y$ | $l_X r_Y$ | $r_X l_Y$ | $r_X r_Y$ |
|---|---|---|---|---|
| $X_G X_B$ | 5, 5 | 5, 5 | 0, 6 | 0, 6 |
| $X_G Y_B$ | 5, 5 | 2, 8 | 3, 3 | 0, 6 |
| $Y_G X_B$ | 5, 5 | 3, 3 | 2, 8 | 0, 6 |
| $Y_G Y_B$ | 5, 5 | 0, 6 | 5, 5 | 0, 6 |

| | $l_X l_Y$ | $l_X r_Y$ | $r_X l_Y$ | $r_X r_Y$ |
|---|---|---|---|---|
| $X_G X_B$ | 0 | 1/4 | 0 | 0 |
| $X_G Y_B$ | 0 | 1/4 | 0 | 0 |
| $Y_G X_B$ | 0 | 0 | 1/4 | 0 |
| $Y_G Y_B$ | 0 | 0 | 1/4 | 0 |

*However, we can observe that this EFCE is not a linear-deviation correlated equilibrium because, for example, Player 1 can increase their payoff by a value of $3/2$ using the following transformation:*

$$X_G X_B \mapsto X_G X_B \qquad \text{\textit{i.e., map the reduced-normal-form plan}} \ (1,0,1,0) \mapsto (1,0,1,0)$$
$$X_G Y_B \mapsto X_G X_B \qquad \text{\textit{i.e., map the reduced-normal-form plan}} \ (1,0,0,1) \mapsto (1,0,1,0)$$
$$Y_G X_B \mapsto Y_G Y_B \qquad \text{\textit{i.e., map the reduced-normal-form plan}} \ (0,1,1,0) \mapsto (0,1,0,1)$$
$$Y_G Y_B \mapsto Y_G Y_B \qquad \text{\textit{i.e., map the reduced-normal-form plan}} \ (0,1,0,1) \mapsto (0,1,0,1)$$

*(Above, we have implicitly assumed that the strategy vectors encode probability of actions in the arbitrary order $X_G$, $Y_G$, $X_B$, $Y_B$). The above transformation is linear, since it can be represented via the matrix*

$$\begin{pmatrix} 1 & 0 & 0 & 0 \\ 0 & 1 & 0 & 0 \\ 1 & 0 & 0 & 0 \\ 0 & 1 & 0 & 0 \end{pmatrix}.$$

*This would swap the pure strategy $X_G Y_B$ with $X_G X_B$ and strategy $Y_G X_B$ with $Y_G Y_B$. Crucially, the strategy at the subtree of information set $B$ is determined by the strategy at the subtree of information set $G$. Thus, the transformed value for each sequence does not purely depend on the ancestors of that sequence, but can also depend on strategies belonging to "sibling" subtrees. Hence, this example proves that LCE $\neq$ EFCE.*

*In fact, we can even verify that in this toy example the set of LCE coincides with the set of all normal-form CE. We refer the interested reader to von Stengel and Forges [2008] for a more detailed discussion of the different behaviors that players can exhibit at an EFCE vs a normal-form CE, which in this specific example also corresponds to a comparison between the EFCE and the LCE.*

**Remark E.2.** *The linear transformation given in Example E.1 also serves to show that the Behavioral Deviations defined in Morrill et al. [2021] are not a superset of all linear transformations. Additionally, behavioral deviations are not a subset of linear transformations, as the latter act only on reduced strategies. Thus the two sets of deviations are incomparable.*

**Example E.3** (LCE $\neq$ CE). *This example was found through computational search. Consider the 2-player game with the following game tree:*

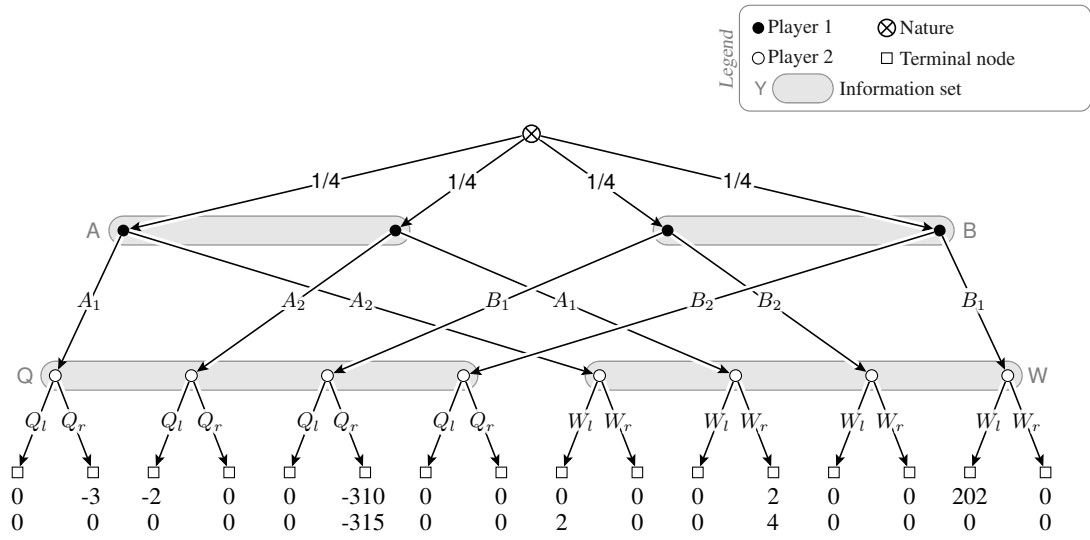

*Based on the previous game tree, the normal-form payoff matrix of the game is shown below.*

| 1 ╲ 2 | $Q_lW_l$ | $Q_lW_r$ | $Q_rW_l$ | $Q_rW_r$ |
|---|---|---|---|---|
| $A_1B_1$ | 50.5, 0.0 | 50.5, 0.25 | -27.75, -78.75 | -27.75, -78.5 |
| $A_1B_2$ | 0.0, 0.0 | 0.5, 1.0 | -0.75, 0.0 | -0.25, 1.0 |
| $A_2B_1$ | 50.0, 0.5 | 49.5, -0.75 | -27.0, -78.25 | -27.5, -79.5 |
| $A_2B_2$ | -0.5, 0.5 | -0.5, 0.0 | 0.0, 0.5 | 0.0, 0.0 |

*We can now verify that the following is a linear-deviation correlated equilibrium for this game. The verifcation can be done computationally by expressing all constraints of Theorem 3.1 as a Linear Program.*

| | $Q_lW_l$ | $Q_lW_r$ | $Q_rW_l$ | $Q_rW_r$ |
|---|---|---|---|---|
| $A_1B_1$ | 1/5 | 0 | 0 | 0 |
| $A_1B_2$ | 0 | 1/5 | 0 | 0 |
| $A_2B_1$ | 1/5 | 0 | 0 | 0 |
| $A_2B_2$ | 0 | 0 | 1/5 | 1/5 |

*However, the swap $\{A_1B_2 \mapsto A_1B_1, A_2B_1 \mapsto A_1B_1\}$ can increase player 1's payoff by $50.5$.*

*Furthermore, we can verify that this swap is not linear as follows. First assume that it was linear and could be written as a matrix $\mathbf{A} \in [0,1]^{\Sigma \times \Sigma}$. Then the matrix has to be consistent with the following transformations:*

$$A_1B_1 \mapsto A_1B_1$$
$$A_1B_2 \mapsto A_1B_1$$
$$A_2B_1 \mapsto A_1B_1$$
$$A_2B_2 \mapsto A_2B_2$$

*For convenience of referring to matrix rows and columns we number the four sequences as:*

$$A_1 : 0, \quad A_2 : 1, \quad B_1 : 2, \quad B_2 : 3$$

*If we focus on the second transformation, $A_1B_2 \mapsto A_1B_1$, we conclude that $\mathbf{A}[:,0] + \mathbf{A}[:,3] = (1,0,1,0)^\top$ which implies that $\mathbf{A}[1,0] = \mathbf{A}[3,0] = \mathbf{A}[1,3] = \mathbf{A}[3,3] = 0$. Now, if we subtract the respective equations of the last two swaps we get $\mathbf{A}[:,2] - \mathbf{A}[:,3] = (1,-1,1,-1)^\top$. Since $\mathbf{A} \in [0,1]^{\Sigma \times \Sigma}$, the last equation implies $\mathbf{A}[1,3] = 1$ which contradicts the previous constraint of $\mathbf{A}[1,3] = 0$. Thus, we have found a valid normal-form swap that cannot be expressed as a linear transformation of sequence-form strategies. Hence, this example proves that LCE $\neq$ CE.*

## F  Details on Empirical Evaluation

In this section we provide details about the implementation of our algorithm, as well as the compute resources and game instances used.

**Implementation of our no-linear-swap-regret dynamics**  We implemented our no-linear-swap-regret algorithm (Algorithm 1) in the C++ programming language using the Gurobi commercial optimization solver [Gurobi Optimization, LLC, 2023], version 10. We use Gurobi for the following purposes.

- To compute the projection needed on Line 5 of Algorithm 1. We remark that while Gurobi is typically recognized as a linear and integer linear programming solver, modern versions include tuned code for convex quadratic programming. In particular, we used the barrier algorithm to compute the Euclidean projections onto the polytope $\mathcal{M}_{\mathcal{Q}\to\mathcal{Q}}$ required at every iteration of our algorithm.
- To compute the fixed points of the matrices $\mathbf{A} \in \mathcal{M}_{\mathcal{Q}\to\mathcal{Q}}$, that is, finding $\mathcal{Q} \ni \boldsymbol{x} = \mathbf{A}\boldsymbol{x}$. As discussed in Section 3 this is a polynomially-sized linear program.
- To measure the linear-swap regret incurred after any $T$ iterations, which is plotted on the y-axes of Figure 2. This corresponds to solving the linear optimization problem

$$\max_{\mathbf{A}\in\mathcal{M}_{\mathcal{Q}\to\mathcal{Q}}} \left\{ \frac{1}{T}\sum_{t=1}^{T} \langle \boldsymbol{\ell}^{(t)}, \boldsymbol{x}^{(t)} - \mathbf{A}\boldsymbol{x}^{(t)} \rangle \right\}.$$

We did very minimal tuning of the constant learning rate $\eta$ used for online projected gradient descent, trying values $\eta \in \{0.05, 0.1, 0.5\}$ (we remark that a constant value of $\eta \approx 1/\sqrt{T}$ is theoretically sound). We found that $\eta = 0.1$, which is used in the plots of Figure 2, performed best.

**Implementation of no-trigger-regret dynamics**  We implemented the no-trigger-regret algorithm of Farina et al. [2022] in the C++ programming language. In this case, there is no need to use Gurobi, since, as the original authors show, the polytope of trigger deviation functions admits a convenient combinatorial characterization that enables us to sidestep linear programming. Rather, we implemented the algorithm and the computation of the trigger regret directly leveraging the combinatorial structure.

**Computational resources used**  Minimal computational resources were used. All code ran on a personal laptop for roughly 12 hours.

**Game instance used**  We ran our code on the standard benchmark game of Kuhn poker Kuhn [1950]. We used a three-player variant of the game. Compared to the original game, which only considers a simplified deck make of cards out of only three possible ranks (Jack, Queen, or Kind), we use a full deck of 13 possible card ranks. The game has 156 information sets, 315 sequences, and 22308 terminal states.

## G  Further Remarks on the Reduction from No-$\Phi$-Regret to External Regret

A no-$\Phi$-regret algorithm is typically defined as outputting *deterministic* behavior from a finite set $\mathcal{X}$ (for example, deterministic reduced-normal-form plans in extensive-form games, or actions in normal-form games), which can be potentially sampled at random. This is the setting used by, for example, Hart and Mas-Colell [2000] and Farina et al. [2022]. However, we remark that when the transformations $\phi \in \Phi$ are linear, any such device can be constructed starting from an algorithm that outputs points $\boldsymbol{x}' \in \mathcal{X}' \coloneqq \text{conv}(\mathcal{X})$, and then sampling $\boldsymbol{x}$ unbiasedly in accordance with $\boldsymbol{x}'$, that is, so that $\mathbb{E}[\boldsymbol{x}] = \boldsymbol{x}'$. The reason why this is useful is that constructing the latter object is usually simpler, as $\mathcal{X}'$ is a closed and convex set, and is therefore amenable to the wide array of online optimization techniques that have been developed over the years.

More formally, let $\Phi$ be a set of linear transformations that map $\mathcal{X}$ to itself. A no-$\Phi$-regret algorithm for $\mathcal{X}' = \text{conv}(\mathcal{X})$ guarantees that, no matter the sequence of loss vectors $\boldsymbol{\ell}^{(t)}$,

$$R'^{(T)} = \max_{\phi\in\Phi} \sum_{t=1}^{T} \langle \boldsymbol{\ell}^{(t)}, \boldsymbol{x}'^{(t)} - \phi(\boldsymbol{x}'^{(t)}) \rangle$$

grows sublinearly. Consider now an algorithm that, after receiving $\boldsymbol{x}'^{(t)} \in \text{conv}(\mathcal{X})$, samples $\boldsymbol{x}^{(t)} \in \mathcal{X}$ unbiasedly, that is, so that $\mathbb{E}[\boldsymbol{x}^{(t)}] = \boldsymbol{x}'^{(t)}$. Then, by linearity of the transformations and

using the Azuma-Hoeffind concentration inequality, we obtain that for any $\epsilon > 0$,

$$\mathbb{P}\left[\left|R'^{(T)} - \max_{\phi \in \Phi} \sum_{t=1}^{T} \langle \boldsymbol{\ell}^{(t)}, \boldsymbol{x}^{(t)} - \phi(\boldsymbol{x}^{(t)}) \rangle\right| \le \Theta\left(\sqrt{T \log \frac{1}{\epsilon}}\right)\right] \ge 1 - \epsilon,$$

where the big-theta notation hides constants that depend on the payoff range of the game and the diameter of $\mathcal{X}$ (a polynomial quantity in the game tree size). This shows that as long as the regret of the no-$\Phi$-algorithm that operates over $\mathcal{X}'$ is sublinear, then so is that of an algorithm that outputs points on $\mathcal{X}$ by sampling unbiasedly from $\mathcal{X}$. We refer the interested reader to Section 4.2 ("From Deterministic to Mixed Strategies") of Farina et al. [2022].

