# OpenReview forum: "Polynomial-Time Linear-Swap Regret Minimization in Imperfect-Information Sequential Games"
_NeurIPS.cc/2023/Conference — NeurIPS 2023 poster_

### Official Review · Reviewer_Dcee · 2023-07-01

**Soundness:** 4 excellent
**Presentation:** 4 excellent
**Contribution:** 3 good
**Rating:** 7
**Confidence:** 4

**Summary:**

Authors introduce a new class of correlated equilibria called linear-deviation correlated equilibria, which can be approached efficiently if all players attain sublinear linear-swap regret.
They show LCEs are distinct from correlated equilibria and extensive-form correlated equilibria and the hardness of maximizing social welfare in LCEs. Finally, they show the difference in no-linear-swap-regret dynamics and no-trigger-regret dynamics to support their theoretical results on a small game.

**Strengths:**

- Introduction of new correlated equilibrium class.
- Technical contribution of a polynomial characterization of the set of all linear transformations from a sequence-form strategy polytope to itself.
- General algorithm for minimizing linear-swap regret.
- Shows relation to existing equilibrium classes.
- Very clearly written paper that I expect will be built upon in the future.


**Weaknesses:**

- This is a theoretical paper and it is not clear what would be the motivation to approach LCEs in practice, compared to another correlation classes.

**Questions:**

- I think it would be of great pedagogical value to make a visual depiction of the different classes of equilibria similar to https://www.cs.cmu.edu/~ggordon/CE/ -- but I understand this might not be possible because the high dimensionality of the joint strategy space.
- Do you think there exists a linear program to find LCEs? If not, why?


**Limitations:**

Limitations are adequately addressed.

---

> ### Author Rebuttal · Authors · 2023-08-09
>
> We thank the reviewer for the comments and observations on our work. Below we address the questions raised and the discussed weaknesses:
>
> (Q1) Thank you, that would indeed be interesting. As you hinted, one likely key difficulty in constructing such a representation is the fact that interesting sequential games (even when extremely simple) tend to have a relatively large number of strategy profiles, which might make a low-dimensional visualization impossible. We remark that focusing on games with sequential moves is necessary in this case, as the special case of normal-form games is not interesting enough since there the EFCE, LCE and CE all coincide. However, we will try to see what we can do for the final version.
>
> (Q2) We agree that this is a very interesting question, to which we also alluded in the future work section (Section 5). In particular, we suspect that it might be possible to devise an efficient centralized algorithm making clever use of LPs in the spirit of Ellipsoid Against Hope that is able to compute one LCE. That is, an algorithm that computes some valid LCE which, however, is not guaranteed to be optimal in any sense. On the other hand, using an LP to *optimize* over the set of LCEs is impossible due to complexity barriers (Thm 4.1).
>
> (W1) Our goal in this paper is to make progress on the challenging question of what is the strongest notion of rationality that can be efficiently attained in EFGs. We believe that the linear-deviation correlated equilibrium (LCE) is worth defining because it emerges from these no-linear-swap regret dynamics, which constitute a natural class of learning dynamics.
>
> An interesting recent development by Mansour et al. [1] is that all notions of rationality weaker than ours automatically allow the environment to exploit the agent. We believe that the play of higher-rationality no-regret learning agents is intrinsically interesting. LCE is the name of the equilibrium points reached by such agents. We expect that future work will analyze LCE’s game-theoretic properties beyond the online learning-related properties already uncovered by us and Mansour et al.
>
> [1] Yishay Mansour, Mehryar Mohri, Jon Schneider, and Balasubramanian Sivan, 2022. Strategizing against learners in bayesian games. COLT.

---

> > ### Comment · Reviewer_Dcee · 2023-08-16
> >
> > Thank you for further enlightening on the topic! Especially for the reference to the new recent work, that is interesting.

---

### Official Review · Reviewer_qFDY · 2023-07-02

**Soundness:** 3 good
**Presentation:** 3 good
**Contribution:** 2 fair
**Rating:** 6
**Confidence:** 4

**Summary:**

This paper studies the convergence of uncoupled strategies to a weakened notion of equilibria called "linear-deviation correlated equilibrium". This equilibrium is reached when all players minimise the no-linear-swap regret which is a specialisation of Phi-equilibria when the set of deviations Phi is the set of all linear transformation from the set (mixed) strategy to itself.
The authors provide efficient implementation of such no-linear-swap regret algorithms and prove that the set of linear-deviation correlated equilibria is a subset of extensive form correlated equilibria. The technical tool used is the of of Gordon 2008 which converts a regret minimiser for the space of deviations to a regret minimiser for the strategy set.

**Strengths:**

The paper aims to solve an important problem, which is the complexity of CE in EFG.
The paper is easy to read, clear and well written and the results seems non-trivial

**Weaknesses:**

The notion of LCE is not well motivated. The authors prove that LCE are EFCE but, while EFCE are well motivated in EFG, LCE do not seems to have any practical meaning. Also the algorithm presented in the paper is slower at converging to EFCEs then the one specifically designed for them (Figure 2 left) and thus do not contribuite in this regard.
The main weakness is that LCE are not an interesting equilibrium to converge to, except from their connection to EFCEs. The author should motivate the introduction of this new equilibria for example by finding examples in which LCE are a reasonable solution concept while other EFCEs are not. Without this the point the paper seems a technical exercise.

**Questions:**

1) Th 31 seems to be the main technical statement of the paper. What are the main challenges of proving this statement? A better intuition of its implications/design would be appriciated
2) Motivate LCE (see Weaknesses)
3) Figure 2 only proves that the inclusions of CE, LCE and EFCE are strict, but is somewhat uninteresting once we have a theorem stating it. What happens is in Figure 2 left you put there time instead of iterations? If the no-linear-swap regret algorithm runs faster then the one that minimises only the trigger-regret this would be a better algorithm for finding EFCEs.

**Limitations:**

yes

---

> ### Author Rebuttal · Authors · 2023-08-09
>
> We thank the reviewer for the comments and observations on our work. Below we address the 3 questions raised:
>
> (Q1) Thanks for the feedback. As mentioned to Reviewer vZdR, we will use the extra content page to revise our paper and include a more detailed intuition of its main proof ideas. We include a high-level, intuitive description of the theorem and the challenges arising in its proof in our response to Reviewer vZdR.
>
> (Q2) We disagree that LCE is interesting only in relation to EFCE. No-trigger-regret agents are robust (hindsight rational) only to trigger deviations, which are a measure-zero set of linear transformations. The question as to what is the highest notion of regret that can be minimized efficiently in imperfect-information sequential games is natural and established. An interesting recent development by Mansour et al. [1] is that all notions of rationality weaker than ours automatically allow the environment to exploit the agent. We believe that the play of higher-rationality no-regret learning agents is intrinsically interesting. LCE is the name of the equilibrium points reached by such agents. We expect that future work will analyze LCE’s game-theoretic properties beyond the online learning-related properties already uncovered by us and Mansour et al.
>
> [1] Yishay Mansour, Mehryar Mohri, Jon Schneider, and Balasubramanian Sivan, 2022. Strategizing against learners in bayesian games. COLT.
>
> (Q3) One thing we wanted to demonstrate with Figure 2 is that in practice there is indeed a perceptible difference between the regret achieved by no-swap and no-trigger regret dynamics, even in fairly small and simple games.
>
> Our no-linear-swap-regret algorithm can be used to minimize any kind of regret that linear-swap deviations subsume (such as the trigger regret). However, the per-iteration time complexity of our linear-swap regret minimization algorithm is worse than that of the trigger-regret minimization algorithm, due to the projection step and the explicit computation of the fixed point (steps (5) and (6) of Algorithm 1). While the algorithm that we provide in the paper (based on the ellipsoid method) is enough for showing that minimizing linear-swap regret can be done in polynomial-time, we suspect that a better understanding of the geometry of the linear deviations polytope could enable sidestepping the expensive ellipsoid method used in our projection step and lead to provably and practically better running times. We leave this interesting question for future work.

---

> > ### Comment · Reviewer_qFDY · 2023-08-15
> >
> > I thank the authors for taking the time of answering my questions. I now understand better the questions that the paper tires to answer. However I'm confused about the arguments regarding [1]. Are the authors claiming that their algorithm can minimise linear swap regret also in bayesian games analysed in [1]?
> >
> > Moreover I still believe that while a general analysis of LCE can be deferred to future works, I think that a brief analysis of their features would greatly improve the paper. In particular a toy example in which EFCEs and LCEs leads to substantially different behaviour would be enough.
> >
> > I now increase my score. Depending on the authors answers I'll be willing to increase further my valuation.
> >
> > [1] Yishay Mansour, Mehryar Mohri, Jon Schneider, and Balasubramanian Sivan, 2022. Strategizing against learners in bayesian games. COLT.

---

> > > ### Author Response · Authors · 2023-08-18
> > >
> > > Thank you a lot for engaging in this discussion and for providing continuous feedback on our paper.
> > > Indeed, our algorithm can minimize linear-swap regret in Bayesian games with a finite set of player types $\Theta$. This follows from the fact that extensive-form games are general enough to capture Bayesian games as well. In particular, we can represent any Bayesian game by using a game tree whose root is a chance node, which randomly initializes the player type to be one of the available $|\Theta|$ types. Information sets are set up so that each player can distinguish only their type and not the other players’. We have also included a short discussion of this in lines 118-123 in the Introduction of our submitted paper pdf.
> > >
> > > Regarding the brief analysis of the features of the LCE, we believe that the Example E.1 from the appendix, used to prove that LCE $\neq$ EFCE, is an interesting toy example showcasing the difference in behaviors that can be exhibited by these equilibria. In this case it happens that the linear deviations capture all possible deviations and, thus, LCE = CE. The game given in the example is a classic instance of a signaling game that has been extensively discussed in the past (eg. see von Stengel and Forges [2008] for an extensive discussion of the different behaviors that are possible for the EFCE and CE, which in our case is equal to the LCE). We will incorporate this discussion in the final version.
> > >
> > > [1] B. von Stengel and F. Forges. Extensive-form correlated equilibrium: Definition and computational complexity. Mathematics of Operations Research, 2008.

---

### Official Review · Reviewer_vZdR · 2023-07-05

**Soundness:** 3 good
**Presentation:** 3 good
**Contribution:** 3 good
**Rating:** 6
**Confidence:** 2

**Summary:**

The paper studies regret minimization in extensive-form games (EFG). Specifically, they study a notion of regret called linear-swap regret, which measures the regret against linear transformations of the player's sequence-form strategies. This notion is stronger than trigger regret (as trigger deviations can be described as a linear transformation of sequence-form strategies) but weaker than swap regret.

The main technical contribution lies in Theorem 3.1, which states that the set of linear transformations (from the set of sequence-form strategies to itself) is a compact polytope defined by a polynomial (in the game tree size) number of linear constraints.

With that in hand, they leverage the template of Gordon et al. [2008] that produces a $\Phi$-regret minimization algorithm given a no-external-regret minimizer $\Phi$ and a fixed point oracle which returns fixed point of transformations in $\Phi$. In this context, the $\Phi$-regret minimization can be chosen, for example, to be Online Gradient Decent; and the fixed point oracle can be implemented using a Linear Programming Solver (e.g., Ellipsoid algorithm). This results in an efficient algorithm that achieves $\sqrt{T}$ linear-swap regret; which in turn also implies an efficient method for finding the correlated equilibrium that corresponds to linear transformations, that the authors call linear-deviation correlated equilibrium (LCE).

Finally, the authors provide examples to show that the set of LCEs strictly contain**s** the set of CEs, and is strictly contain**ed** in the set of Extensive-form correlated equilibrium (EFCE). They also show that the set of Behavioral correlated equilibrium (BCE) and the set of LCEs are incomparable.



**Strengths:**

The paper is written well; the authors give great background on EFGs, on different notions of correlated equilibrium, and their relation to different notions of regret.
The construction of the algorithm and results are fairly easy to understand, and the authors express well the difference between their results and previous work by providing examples that compare their notion of equilibrium with existing notions.
Finally, I agree that determining *"what is the strongest notion of rationality that can be attained efficiently"* is an interesting open problem, and indeed the paper provide some progress in this direction.

**Weaknesses:**

- There is no obvious algorithmic contribution here: As noted, Theorem 3.1 is the main technical contribution which is purely geometric. Given that, constructing the algorithm is fairly straightforward with the scheme of Gordon et al. [2008].
- While the background the paper gives is quite solid, it is not very proportional to the relatively small amount of attention the authors give to their technical contribution. In particular, while the statement of Theorem 3.1 is clear, it is not very intuitive, and it is hard to understand why it is correct. This is also the main reason that my confidence score is relatively low. I think that some of the preliminaries, as well as the empirical evaluations, could be deferred to the appendix - instead, I think that giving the main proof ideas is more important.

**Missing related work:** I think that the work of Anagnostides et al. (2022) is very relevant here and motivates important questions for future work. Specifically, this work achieves a much better rate of $\log T$ for trigger regret. Thus, it is fairly natural to ask whether this is also attainable for linear-swap regret, especially since they also built upon the template of Gordon et al. [2008].

Anagnostides, I., Farina, G., & Sandholm, T. (2022). Near-Optimal $\Phi $-Regret Learning in Extensive-Form Games. arXiv preprint arXiv:2208.09747.‏

**Questions:**

N/A

---

> ### Author Rebuttal · Authors · 2023-08-09
>
> We thank the reviewer for the comments and observations on our work. Below we address the weaknesses discussed:
>
> (W1) The geometric contribution is the key to the algorithmic contribution of the paper: the structure provided by our characterization theorem (Thm 3.1) is what enables constructing agents minimizing regret with respect to all linear deviations in polynomial time in imperfect-information sequential games. Hence, one cannot untangle the two: without the geometric contribution, there is no algorithmic contribution.
>
> As correctly mentioned in the review, Gordon et al. [2008] provide an elegant recipe for constructing a no-$\Phi$-regret algorithm, as long as two key ingredients can be provided: a polynomial-time (per iteration) no-external-regret algorithm for the set of deviations $\Phi$, and an efficient fixed point oracle. The first of these components can be rather complicated, as is the case in our paper, since the structure of the set of deviation functions $\Phi$ can be arbitrarily complex.
>
>
> (W2) Thank you for proposing this improvement in our paper presentation. Since the camera ready allows an extra content page, we will use that to include a more detailed intuition of its main proof ideas. We include here below a more high-level, intuitive description of the theorem and the challenges arising in its proof.
>
> The proof proceeds by induction on the game tree as follows:
>
> * The Base Case (appendix, pg. 15) corresponds to the case of being at a leaf decision point, that is, an information set for which all actions lead to termination of the game. In this case, the set of deviations corresponds to all linear transformations from a probability $n$-simplex into a given polytope $\mathcal{P}$ in $\mathbb{R}^d$. This set is equivalent to all $d$ x $n$ matrices whose columns are points in $\mathcal{P}$, which is not hard to verify mathematically. This corresponds to constraint (3) of our characterization.
>
> * For the inductive step, we are at an intermediate decision point (information set of the learning player), that is, one for which at least one action leads to further decision points. Let j be the decision point.
>
>   In Lemma B.8 we show that any terminal action a at j (that is, such that no further decision points follow under that) leads to a column in the transformation matrix that is necessarily a valid point in the polytope $\mathcal{P}$. This is similar to the base case, and leads to constraint (3) in our formulation.
>
>   In Lemma B.7, we look at the other case of a nonterminal action a at j (that is, such that one or more further decision points follow under that) is such that the corresponding column in the transformation matrix is identically 0 (constraint (4)) without loss of generality. This allows for the "crux" of the transformation to happen in the subtree rooted at ja. A key difficulty is in characterizing all valid transformations at such a subtree. In particular, several decision points can have ja as their parent sequence. The set of strategies in the subtree rooted at ja is therefore in general the Cartesian product of subtrees rooted at each of the decision points whose parent sequence is ja. This intuitively explains the need for the (fairly technical and involved) Proposition B.4, whose goal is to precisely characterize valid transformations of Cartesian products. The characterization for the Cartesian products leads to constraints (5) and (6) in our final characterization.
>
>
> (Missing related work) Thank you for the intriguing question, we will definitely include a discussion about that paper in the revision. The question as to whether $O(\log T)$ regret per-player can be attained in self-play is very interesting. While the paper you mentioned proposes a general methodology that applies to CEs in normal-form games and EFCE/EFCCE in sequential games, the authors’ construction fundamentally relies on being able to express the fixed points computed by the algorithm as (linear combinations of) rational functions with positive coefficients of the deviation matrices. In the case of CEs, this characterization follows from the Markov chain tree theorem, while in EFCE and EFCCE this fundamentally follows from the fact that the fixed points can be computed inductively, solving for stationary distributions of local Markov chains at each decision point. In the case of linear transformations considered in our paper, such a local characterization of the fixed point is unknown. We will definitely include a discussion about this, thanks again for the suggestion.

---

> > ### Comment · Reviewer_vZdR · 2023-08-13
> >
> > I thank the authors for their response - I have no further questions at this point.

---

### Official Review · Reviewer_V8FX · 2023-07-14

**Soundness:** 3 good
**Presentation:** 3 good
**Contribution:** 3 good
**Rating:** 6
**Confidence:** 4

**Summary:**

This paper focuses on addressing the challenge of minimizing linear swap regret in extensive form games, which is considered a stronger notion compared to trigger regret in Extensive Form Correlated Equilibrium (EFCE). To achieve efficient implementation of the Phi-regret minimization problem (where Phi set contains all linear swap transformation), the authors provide a polynomial-sized description of all linear swap operators. This result shows that there exists a no linear swap regret algorithm with polynomial-time iteration complexity based on the size of the game tree. Additionally, the paper presents experimental results that illustrate how the equilibria achieved through the no linear swap regret algorithm exhibit greater strength compared to EFCE.

**Strengths:**

This paper introduces a novel and essential contribution by providing a polynomial characterization of the set of all linear transformations from a sequence-form strategy polytope to itself. This characterization holds significant value for future research on linear-swap-regret.




**Weaknesses:**

The contribution of this paper may be perceived as limited. The primary and significant contribution lies in the polynomial description of the linear swap transformation set. Given the sequential structure of the sequence form strategy, this finding is not entirely unexpected.

It appears that the paper lacks some intuitive explanations regarding the concept of linear swap (further details can be found in the questions raised). There is a sense that LCE (Linear Swap Correlated Equilibrium) is merely one equilibrium between CE (Correlated Equilibrium) and EFCE (Extensive Form Correlated Equilibrium), which is relatively easy to compute. The experimental results may not comprehensively demonstrate the potential and strength of LCE



**Questions:**

1. Is there any intuition behind the linear swap transformation, apart from its linearity? For instance, can LCE be interpreted with the presence of a mediator? Why do you believe LCE is an equilibrium worth considering?

2. Maximizing social welfare is known to be challenging. However, in your experiments, did you observe any significant differences in social welfare between No-linear-swap-regret dynamics and No-trigger-regret dynamics?

3. Since No-linear-swap-regret dynamics are stronger than No-trigger-regret dynamics. The result in Figure 2 is expected. What about the running times? Is the running time of no-linear-swap-regret dynamics substantially larger?

[1] Zhang, Brian, and Tuomas Sandholm. "Polynomial-time optimal equilibria with a mediator in extensive-form games." Advances in Neural Information Processing Systems 35 (2022): 24851-24863.

**Limitations:**

The authors have partially addressed the limitations of their work, though there is space for improvement (see the section Weaknesses).

---

> ### Author Rebuttal · Authors · 2023-08-09
>
> We thank the reviewer for the comments and observations on our work. Below we address the questions raised.
>
> (Q1) We believe linear-deviation correlated equilibria (LCEs) are most naturally understood as the name of the equilibrium points that emerge from the higher-rationality no-regret learning agents we construct. To recap briefly, our main goal in this paper is to make progress on the challenging question of what is the strongest notion of no-regret learning/hindsight rationality that can be attained in polynomial time in imperfect-information sequential multiagent settings. The best prior results only applied to extremely structured and isolated subsets of linear transformations, including EFCE deviations and communication deviations (exhibiting a large gap compared to what is instead possible in matrix/nonsequential games, where no-swap-regret can be attained efficiently). In our paper, we show that learning to be robust (i.e. not regret) **any** linear transformation can be guaranteed in polynomial time, subsuming virtually all prior known notions (including EFCCE, EFCE, and the very recent work on communication equilibria in Bayesian games by Fujii [1]). Another reason to care about this notion of rationality is that it has been recently shown by Mansour et al. [2] that all weaker notions of rationality automatically allow the environment to exploit the agent.
>
> One can also think of LCE in the context of a mediator (correlation device) that samples strategy profiles and recommends the corresponding pure strategies to players. The mediator’s concern is to find a correlated distribution of play such that no player would be better off by deviating unilaterally using any linear transformation of their strategy. This is akin to, but significantly more general than, EFCE, where the players can only use trigger deviations, a special class of linear transformations. However, while trigger deviations can be re-interpreted as the players being able to transform behavior conditionally only on the recommendation of one information set, we do not know if such an intuitive interpretation can be given about the set of all linear deviations. While this paper is mostly focused on the learning and computational aspects, exploring these game-theoretic modeling questions for LCE is an interesting question for future work.
>
> [1] Kaito Fujii, 2023. Bayes correlated equilibria and no-regret dynamics. Arxiv.
>
> [2] Yishay Mansour, Mehryar Mohri, Jon Schneider, and Balasubramanian Sivan, 2022. Strategizing against learners in bayesian games. COLT.
>
> (Q2) Thanks for the interesting question. Since LCE is a subset of EFCE, the maximum social welfare that can be achieved by LCE can only be $\leq$ of that reachable by EFCE (intuitively, the agents are more rational, so it takes more effort to incentivize them towards any correlated behavior). Indeed, we empirically observe that the utility reached by LCE is lower than that of EFCE in the experiments of Figure 2. You can observe this for example in the experimental data we included in the supplemental material, together with the implementation of our dynamics.
>
> (Q3) One thing we wanted to demonstrate with Figure 2 is that in practice there is indeed a perceptible difference between the regret achieved by no-swap and no-trigger regret dynamics, even in fairly small and simple games.
>
> The per-iteration time complexity of our linear-swap regret minimization algorithm is worse than that of the trigger-regret minimization algorithm, due to the projection step and the explicit computation of the fixed point (steps (5) and (6) of Algorithm 1). While the algorithm that we provide in the paper (based on the ellipsoid method) is enough for showing that minimizing linear-swap regret can be done in polynomial-time, we conjecture that a better understanding of the geometry of the linear deviations polytope could enable sidestepping the expensive ellipsoid method used in our projection step and lead to provably and practically better running times. We leave this interesting question for future work.
>
> (W) Finally, we would like to leave a small comment on the sentence "Given the sequential structure of the sequence form strategy, this finding is not entirely unexpected." from the Weaknesses section. Specifically, we would like to highlight that the proof of this characterization is very far from trivial, as is also evident from the appendix included in the supplemental material.

---

> > ### Comment · Reviewer_V8FX · 2023-08-15
> >
> > Thank you for answering my questions! I have increased my score to 6.

---

### Author Rebuttal · Authors · 2023-08-09

We thank all the reviewers for the detailed comments and constructive feedback. We have addressed the reviewers’ comments/questions individually below.

---

### Decision · Program_Chairs · 2023-09-21

**Decision:**

Accept (poster)

**Comment:**

This paper proposes a new notion of correlated equilibrium called linear-deviation correlated equilibrium (LCE) for extensive-form games (EFGs) that is stricter than the commonly studied EFCE, yet still admits polynomial-time algorithm using an explicit characterization of the strategy deviation set. After rebuttal, all reviewers are positive about the contributions of this paper to the EFG literature. Therefore I recommend acceptance.